https://doi.org/10.1038/s41467-019-08384-x · **OPEN**

# Remodeling of secretory lysosomes during education tunes functional potential in NK cells

Jodie P. Goodridge[1,2], Benedikt Jacobs[1,2], Michelle L. Saetersmoen[1,2], Dennis Clement[1,2], Quirin Hammer [3], Trevor Clancy [1,2], Ellen Skarpen[4], Andreas Brech[4], Johannes Landskron[1,5], Christian Grimm[6], Aline Pfefferle[3], Leonardo Meza-Zepeda [7,8], Susanne Lorenz[8], Merete Thune Wiiger[1,2], William E. Louch[9], Eivind Heggernes Ask [1,2], Lisa L. Liu[3], Vincent Yi Sheng Oei [1,2], Una Kjällquist[10], Sten Linnarsson [10], Sandip Patel[11], Kjetil Taskén[1,2,5], Harald Stenmark[4] & Karl-Johan Malmberg [1,2,3]

Inhibitory signaling during natural killer (NK) cell education translates into increased responsiveness to activation; however, the intracellular mechanism for functional tuning by inhibitory receptors remains unclear. Secretory lysosomes are part of the acidic lysosomal compartment that mediates intracellular signalling in several cell types. Here we show that educated NK cells expressing self-MHC specific inhibitory killer cell immunoglobulin-like receptors (KIR) accumulate granzyme B in dense-core secretory lysosomes that converge close to the centrosome. This discrete morphological phenotype is independent of transcriptional programs that regulate effector function, metabolism and lysosomal biogenesis. Meanwhile, interference of signaling from acidic $Ca^{2+}$ stores in primary NK cells reduces target-specific $Ca^{2+}$-flux, degranulation and cytokine production. Furthermore, inhibition of PI(3,5)$P_2$ synthesis, or genetic silencing of the PI(3,5)$P_2$-regulated lysosomal $Ca^{2+}$-channel TRPML1, leads to increased granzyme B and enhanced functional potential, thereby mimicking the educated state. These results indicate an intrinsic role for lysosomal remodeling in NK cell education.

[1] The KG Jebsen Center for Cancer Immunotherapy, Institute of Clinical Medicine, University of Oslo, 0318 Oslo, Norway. [2] Department of Cancer Immunology, Institute for Cancer Research, Oslo University Hospital, 0310 Oslo, Norway. [3] Center for Infectious Medicine, Department of Medicine Huddinge, Karolinska Institutet, 14186 Stockholm, Sweden. [4] Department of Molecular Cell Biology, Institute for Cancer Research, Oslo University Hospital, 0310 Oslo, Norway. [5] Centre for Molecular Medicine Norway, Nordic EMBL Partnership, University of Oslo and Oslo University Hospital, 0318 Oslo, Norway. [6] Department of Pharmacology and Toxicology, Faculty of Medicine, University of Munich (LMU), Munich 80336, Germany. [7] Department of Tumor Biology, Institute for Cancer Research, The Norwegian Radium Hospital, Oslo University Hospital, Oslo 0310, Norway. [8] Genomics Core Facility, Department of Core Facilities, Institute for Cancer Research, The Norwegian Radium Hospital, Oslo University Hospital, Oslo 0310, Norway. [9] Institute for Experimental Medical Research, Oslo University Hospital and University of Oslo, 0424 Oslo, Norway. [10] Division of Molecular Neurobiology, Department of Medical Biochemistry and Biophysics, Karolinska Institutet, 17177 Stockholm, Sweden. [11] Department of Cell and Developmental Biology, University College London, Gower Street, London WC1E 6BT, UK. Correspondence and requests for materials should be addressed to K.-J.M. (email: k.j.malmberg@medisin.uio.no)

Natural killer (NK) cells achieve specificity through unique combinations of germ-line encoded receptors. These receptors are critical for the development of cell-intrinsic functional potential, enabling spontaneous activation upon recognition of target cells displaying reduced class I MHC expression[1]. Inhibitory interactions with self-MHC translate into a predictable quantitative relationship between self-recognition and effector potential, a process termed NK cell education[2]. Despite being clearly evident in different species[3], NK cell education operates through an as yet largely unknown mechanism. Paradoxically, mature NK cells expressing self-MHC-specific inhibitory receptors, receiving constitutive inhibitory input during homeostasis, exhibit increased levels of functionality upon ligation of activating receptors[2,4].

Mouse models have demonstrated that this functional phenotype is dynamic and dependent on the net signaling input to NK cells during cell-to-cell interactions with both stromal and hematopoietic cells[5]. Transfer of mature NK cells from one MHC environment to another results in reshaping of the functional potential based on the inhibitory input of the new MHC setting[6]. Alternatively, genetic knock-down of SLAM-family receptors by CRISPR/Cas9 leads to hyperfunctionality[7], whereas deletion of the inhibitory signaling through ITIM and SHP-1 renders NK cells hypofunctional[4,8]. However, it remains unclear how and when the net signaling input from activating and inhibitory receptors during NK cell education is integrated to tune the functional potential of the cell.

One difficulty in establishing the cellular and molecular mechanisms that account for the calibration of NK cell function is the lack of a steady-state phenotype that defines the educated NK-cell state. Functional readouts used to distinguish self-specific NK cells from hyporesponsive NK cells do not provide information about the prior events that culminate in the development of effector potential. Apart from differences in the relative levels and distribution of NK cell receptors at the cell membrane[9,10], transcriptional and phenotypic readouts at steady state provide scant differences between self and non-self-specific NK cells[11,12]. Whether inhibitory signaling is converted into a paradoxical gain of function through an as yet unknown mechanism (e.g., arming/stimulatory licensing), or whether expression of self-specific inhibitory receptors protect the cell from tonic activation that would otherwise lead to erosion of function over time (e.g., disarming/inhibitory licensing) remains to be determined[13,14].

Here, we show that expression of self-specific inhibitory receptors influences the structural organization of the endolysosomal compartment. This allows NK cells to sequester granzyme B and mount strong, receptor-triggered effector responses from pre-existing large dense-core secretory lysosomes (also referred to as lytic granules). Moreover, the secretory lysosomes form part of the acidic $Ca^{2+}$ stores in the cells and contribute to the global $Ca^{2+}$-flux and downstream effector function in NK cells. These findings connect homeostatic receptor input to lysosomal homeostasis, which tune the functional potential in self-KIR+ NK cells.

## Results

### Accumulation of granzyme B in educated human NK cells. 
The impact of NK cell education on degranulation of primary NK cells expressing self- versus non-self-specific KIR was examined in 88 healthy blood donors (Fig. 1a). In line with the previous studies, NK cells expressing self-specific KIR exhibited greater degranulation in response to HLA class I-deficient K562 cells. To address the mechanisms involved in the tuning of effector potential, the expression of granzyme B, a core effector molecule, was monitored by flow cytometry in mature NK cells stratified on

the expression of self- versus non-self-specific KIR. The stochastic expression of KIR in NK cells occurs independently of MHC setting, providing unique situation in which self and non-self-specific KIR+ subsets can be examined within each individual as a natural equivalent of gene-silencing[15,16]. This allowed us to address the impact of reciprocal presence or absence of a self-KIR on the total granzyme B content within equivalent subsets in each individual. Extended analysis of 64 healthy donors showed significantly higher expression of granzyme B in NK cells positive for KIR2DL3 (2DL3) relative to KIR2DL1 (2DL1) from individuals homozygous for the 2DL3 ligand, HLA-C1/C1 (Fig. 1b). Conversely, granzyme B was elevated in 2DL1+ cells from individuals homozygous for the 2DL1 ligand, HLA-C2/C2. In order to control for the stage of differentiation, which is known to influence the expression of effector molecules[17], these analyses were performed in NK cells that were NKG2A negative and CD57 negative (Supplementary Figure 1a). Corroborating the link between inhibitory input through self-KIR and granzyme B expression, donors that were heterozygous for HLA-C1/C2 had similarly high levels of granzyme B in both 2DL1 and 2DL3 single-positive NK cells (Fig. 1b).

Accumulation of granzyme B in self-KIR+ NK cells was confirmed in a second cohort of 49 healthy donors and was observed in both KIR A/A and B/x haplotypes, which differ in terms of their content of activating KIR genes (Supplementary Figure 1b-c)[18]. Granzyme B expression was also higher in 3DL1+ NK cells from donors positive for its cognate ligand HLA-Bw4 (Fig. 1c), particularly in those who possessed strong educating motifs, e.g., a Bw4 allotype with isoleucine (Ile) at position 80 whereas granzyme B was lower in NK cells carrying the weak A24 motif alone (Supplementary Figure 2a, b)[19,20]. Several alleles of KIR2DL2 (2DL2) have been shown to bind to both HLA-C1 and HLA-C2[21,22]. In line with such cross-reactive binding patterns, 2DL2/2DS2 single-positive NK cells expressed similar levels of granzyme B in HLA-C1/C1, -C1/C2, and -C2/C2 donors (Fig. 1d). KIR2DS1 binds to HLA-C2 but does not endow NK cells with function and had low levels of granzyme B (Fig. 1e). Finally, it is well established that NKG2A/HLA-E interactions contribute to the education of NK cells[23,24]. In line with the results observed in educating single KIR+ NK cell subsets, NKG2A+KIR−CD57− NK cells expressed higher levels of granzyme B (Fig. 1f).

To study the effect of self KIR expression in a dynamic model, retroviral transduction was used to introduce full-length 2DL1 or 2DL3 into NK cell lines YTS (HLA-C1/C1) and NKL (HLA-C2/C2). Extending the findings with ex vivo staining of primary NK cells, the transduced NK cell lines showed a similar accumulation of granzyme B following transfection of a self KIR (Fig. 1g) and enhanced functionality (Supplementary Figure 3a-c). These data show that the expression of inhibitory self-KIR or NKG2A is associated with an increased granzyme B payload in NK cells, establishing a link between inhibitory input and the regulation of the core cytolytic machinery of NK cells.

**Granzyme B accumulation is independent of transcription.** To address whether the increased levels of granzyme B in educated NK cells were due to increased transcription, NKG2A−NKG2C−CD57− NK cells were sorted into 2DL3 or 2DL1 single-positive populations from C1/C1 and C2/C2 donors, and transcriptionally profiled using RNA-Seq (Supplementary Figure 4a, b). In line with the previous studies in mice[9], there was a near perfect correlation between genes expressed in self and non-self KIR+ human NK cell subsets, including granzyme B and genes encoding transcription factors for effector loci, lysosomal biogenesis, and mechanistic target of rapamycin (mTOR) regulated metabolism (Fig. 2a and Supplementary Table 1). For reference,

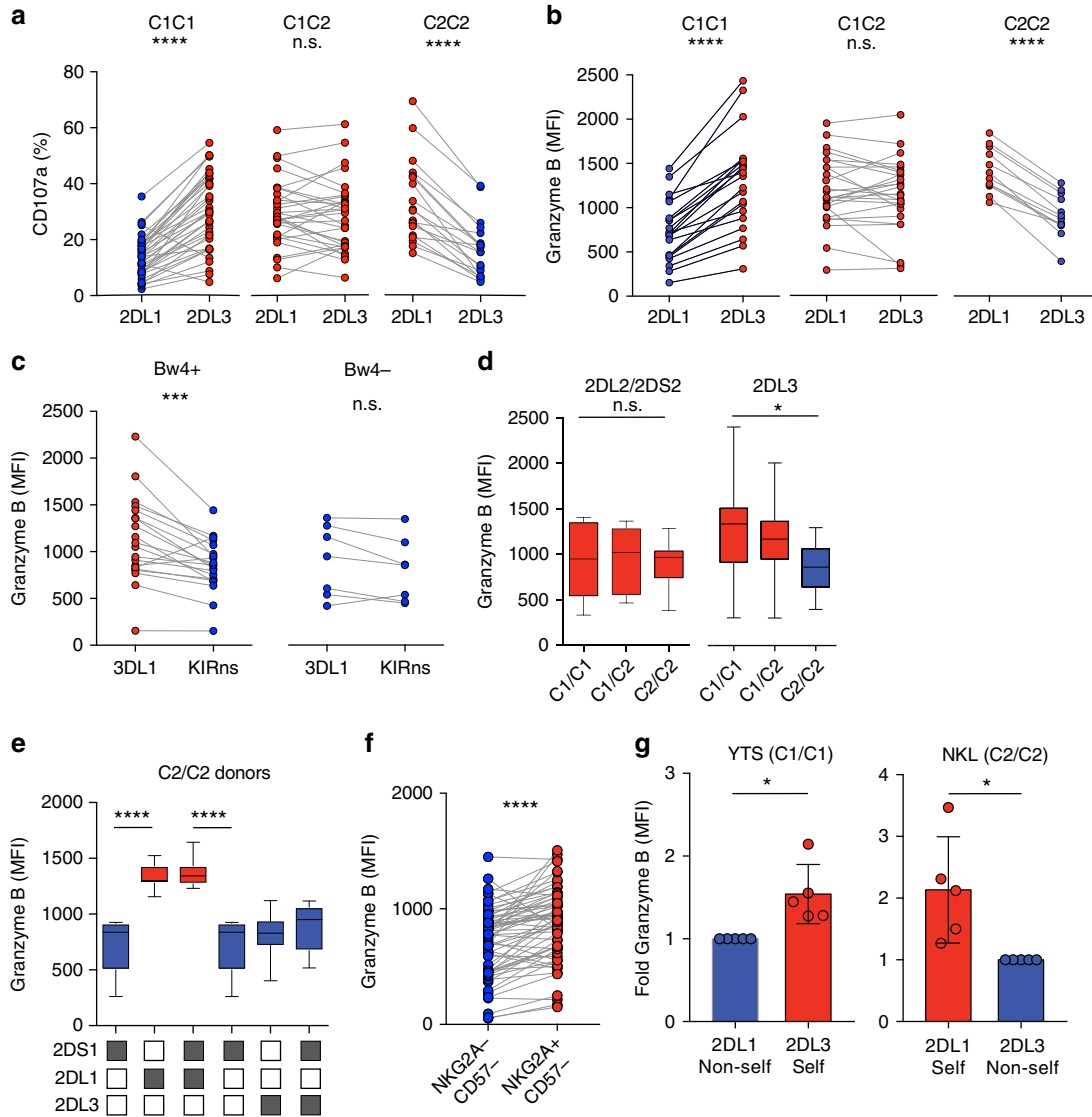

**Fig. 1** NK cell education is associated with accumulation of granzyme B. **a** CD107a expression in response to K562 cells by the indicated KIR subset of resting mature CD56dim NKG2A− NKG2C−CD57− NK cells from C1/C1 (n = 38 donors), C1/C2 (n = 31 donors), and C2/C2 (n = 19 donors). **b** Expression of granzyme B in 2DL3 and 2DL1 single-positive NKG2A−CD57− NK cells from C1/C1 (n = 21 donors), C1/C2 (n = 26 donors), and C2/C2 (n = 12 donors). Donors with less than 100 events in the final KIR gate were excluded. **c** Expression of granzyme B in 3DL1+/− NKG2A−CD57− NK cells from Bw4+ (n = 20 donors) and Bw4− (n = 7 donors). **d** Expression of granzyme B in 2DL2/2DS2 or 2DL3-single-positive NKG2A− NKG2C−CD57− NK cells in C1/C1 (n = 9 donors), C1/C2 (n = 8 donors), and C2/C2 (n = 9 donors). **e** Expression of granzyme B in the indicated NKG2A− NKG2C−CD57− NK cell subset in C2/C2 (n = 7 donors). **f** Expression of granzyme B in KIR−CD57−NKG2A+/− CD57− NK cells (n = 63 donors). **g** Expression of granzyme B in YTS and NKL cells transfected with 2DL3 or 2DL1. The graph shows data from one representative experiment of two. Paired t-tests were performed in (**a**–**c**, **f**–**g**). One-way ANOVA tests followed by Tukey's multiple comparison tests were performed in (**e**, **f**). Whiskers show 5th to 95th percentile. Bars show the median. ****p < 0.0001; ***p < 0.001; and *p < 0.05. Red and blue circles and box plots represent NK cells with self or non-self KIR, respectively

we also sorted and performed RNA-Seq on NK cell subsets at discrete stages of NK cell differentiation CD56bright, mature CD56dimNKG2A−KIR−, and CD56dimNKG2A−KIR+ (Supplementary Figure 5). As expected, NK cell differentiation and KIR acquisition were associated with increased transcription of granzyme B and several other genes known to be involved in regulating effector function, including IRF4 and PRDM1 (Supplementary Figure 5). These results were confirmed in a panel of eight selected qPCR targets comprising transcription factors and canonical cell surface markers linked to NK cell differentiation and showing a marked dissociation between differentiation and education (Fig. 2b and Supplementary Figure 6). Together, these data demonstrated that the increased levels of granzyme B detected by flow cytometry in self-KIR+ NK cells occur

independently of transcriptionally regulated programs, including differences in tonic metabolic input to the cell.

In mouse NK cells, the expression of granzyme B is regulated by cytokine-induced translation from a preexisting pool of mRNA transcript[25]. Therefore, we explored the possibility that self and non-self-specific NK cells may respond differentially to cytokine priming in vivo, resulting in divergent steady-state levels of expressed granzyme B. To address this possibility, NK cells exposed to IL-15 or IL-21 for various lengths of time were monitored for granzyme B content using flow cytometry (Fig. 2c). Both self and non-self KIR+ CD56dim NK cells displayed increased levels of granzyme B in response to IL-15 and IL-21 stimulation. Notably, the relative differences in granzyme B between self and non-self-specific NKG2A−NKG2C−CD57− NK

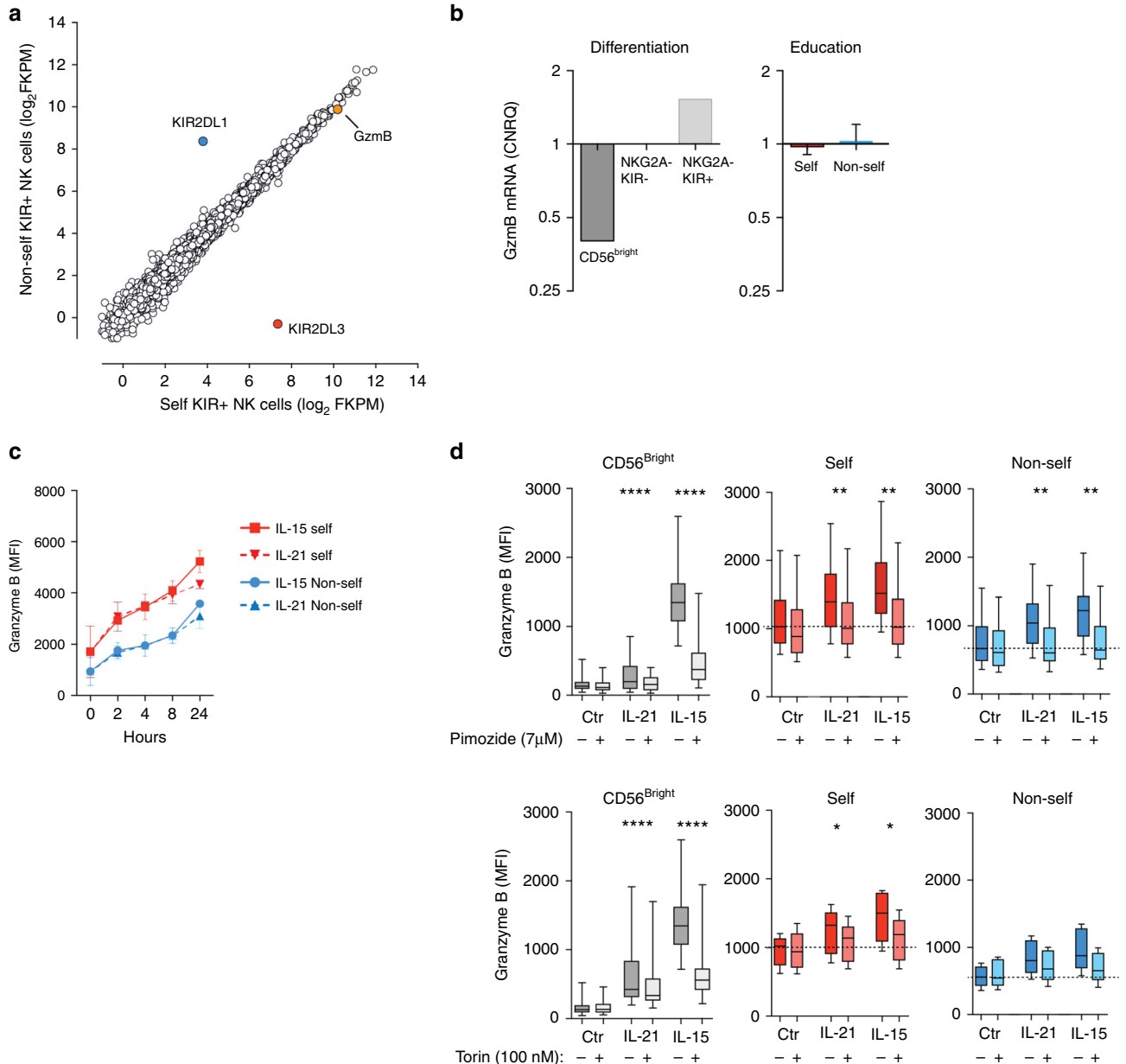

**Fig. 2** Granzyme B accumulation is independent of transcription. **a** Global RNA-Seq of sorted self-KIR$^+$ and non-self-KIR$^+$ NKG2A$^-$ NKG2C$^-$CD57$^-$ NK cells. The figure depicts one representative *C1/C1* donor out of three independent donors. **b** Quantitative PCR of granzyme B mRNA in sorted NK cells at different stages of differentiation (left, 4 pooled samples) and in NKG2A$^-$NKG2C$^-$CD57$^-$ NK cells expressing a self- or non-self KIR (right, $n = 5$ paired samples). **c** Expression of granzyme B in the indicated NK cell subsets following stimulation with IL-15 or IL-21 for the indicated length of time (summary of two independent experiments using two different donors). **d** Expression of granzyme B after 24 h of stimulation with IL-21 or IL-15 in CD56$^{bright}$ NK cells and NKG2A$^-$ NKG2C$^-$CD57$^-$ single-positive CD56$^{dim}$ NK cells expressing a self- or non-self KIR NK cell subset in the presence or absence of the STAT-5 inhibitor Pimozide (top) (7 μM) and the mTOR inhibitor Torin-1 (100 nM) (bottom). The data in panel (**d**) are the summary of at least two independent experiments. Whiskers show 5th to 95th percentile. Bars show the median. Wilcoxon paired, non-parametric tests. ****$p < 0.0001$ and **$p < 0.01$. Red and blue circles, connecting lines and box plots represent NK cells with self or non-self KIR, respectively

cells were similar after stimulation with IL-15 or IL-21 (Fig. 2c). Furthermore, blockade of STAT-5 and mTOR signaling with Pimozide and Torin-1, respectively, abolished the cytokine-induced increase in granzyme B in both self and non-self-specific NK cells (Fig. 2d). These data indicate that a stable pool of granzyme B is retained by NK cells independently of constitutive input through cytokine or metabolic signals.

**Education is associated with remodeling of the lysosomes.** The finding that self-KIR$^+$ NK cells expressed higher levels of

granzyme B independently of mRNA levels provided initial insights into possible post-transcriptional mechanisms underlying the increased functional potential associated with NK cell education. Granzyme B is sequestered into acidic secretory lysosomes[26]. To determine whether the increased levels of granzyme B in self-specific NK cells were the result of higher density, number, or size of such secretory lysosomes, NKG2A$^-$ NKG2C$^-$CD57$^-$ NK cells were sorted into self- or non-self-specific NK cell subsets, imaged by confocal microscopy and analyzed in a blinded fashion. Corroborating the difference in granzyme B expression observed using flow cytometry, self-KIR$^+$ NK cells

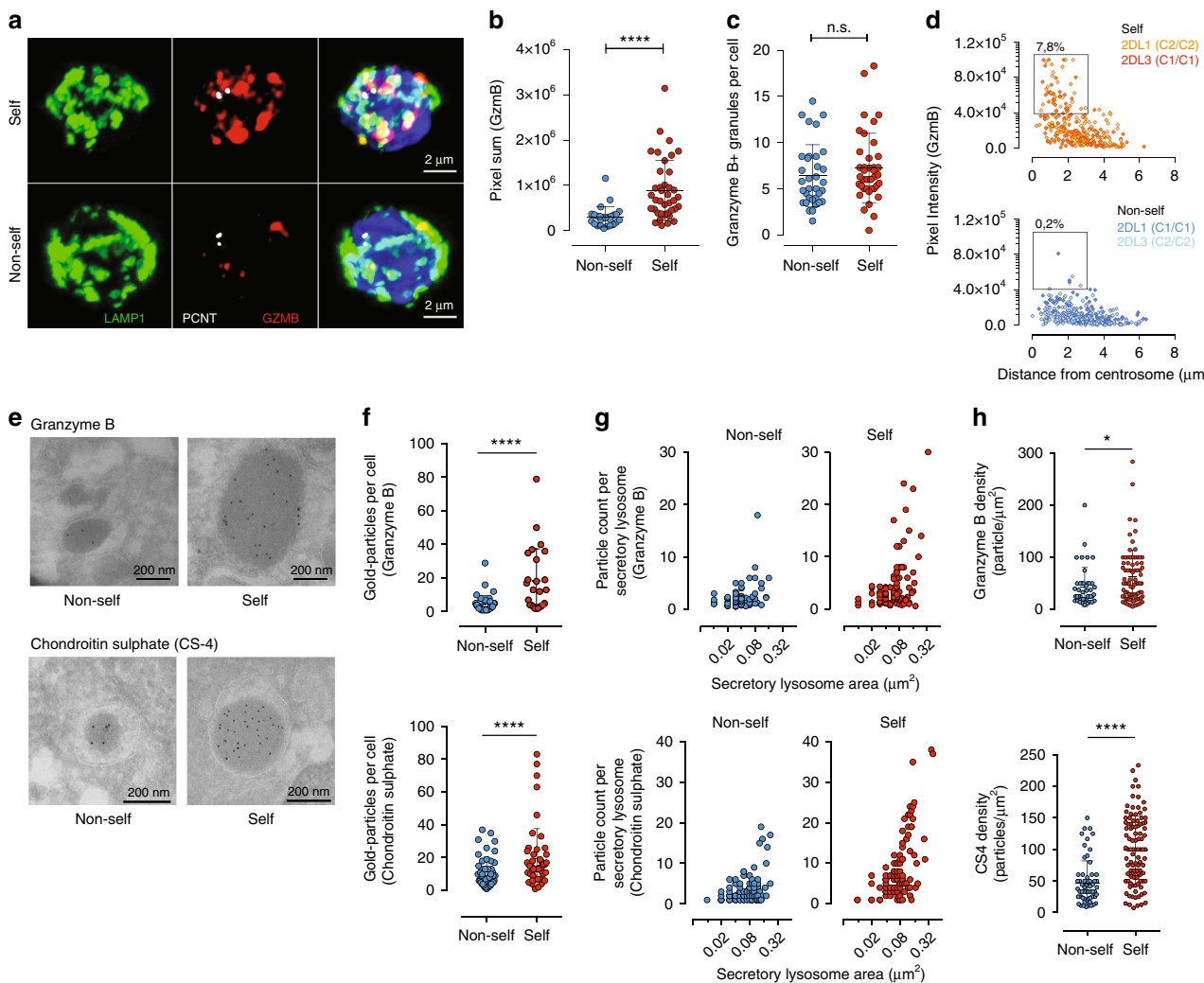

**Fig. 3** Modulation of the lysosomal compartment in educated NK cells. **a** Confocal microscopy Z-stack showing Pericentrin (PCNT), LAMP-1, and granzyme B (GZMB) staining in sorted mature CD56$^{dim}$ NKG2A$^-$NKG2C$^-$CD57$^-$ NK cells expressing non-self or self KIR. Scale bar is 2 μm. **b** The pixel sum of granzyme B staining in cells expressing non-self or self KIR. **c** The number of LAMP-1$^+$ lysosomal structures. Data in panels (**b**) and (**c**) are aggregated from sorted 2DL1 and 2DL3 single-positive NK cell subsets from C1C1 (n = 5 donors) and C2C2 (n = 5 donors) donors. **d** Granzyme B expression levels versus the distance from the centrosome in individual lysosomes in sorted NKG2A$^-$NKG2C$^-$CD57$^-$ NK cells expressing non-self or self KIR (n = 804 lysosomes from 3 donors were analyzed). Gates were set based on visual inspection to quantify the percentage of granzyme-B dense secretory lysosomes in self and non-self NK cells. **e** Representative immuno-EM section showing staining with gold-particle coated anti-granzyme B (top) and Chondroitin Sulphate-4 (CS-4) (bottom) of sorted CD56$^{dim}$ NKG2A$^-$NKG2C$^-$CD57$^-$ NK cells expressing non-self or self KIR. Scale bar is 200 nm. **f** Number of gold particles (granzyme B and CS-4) per cell. (non-self n = 83 cells, self n = 109 cells). **g** Particle count (granzyme B and CS-4) as a function of the lysosomal area. **h** Density of gold particles (granzyme B and CS-4) per lysosomal area (μm$^2$). Immuno-EM data are from 5 donors and 5 experiments. Paired t-tests were performed in panels (**b**, **c**, **e**, and **g**). ****p < 0.0001 and *p < 0.05. Red and blue circles and box plots represent NK cells with self or non-self KIR, respectively

had a higher overall intensity of granzyme B staining (Fig. 3a, b and Supplementary Figure 7a). The number of secretory lysosomes in the two NK cell subsets, however, was not changed (Fig. 3c). Notably, the most intense labeling was found at a close distance to the centrosome in self-KIR$^+$ NK cells (Fig. 3d and Supplementary Figure 7b). These granzyme-B dense secretory lysosomes, localized close to the centrosome were uniquely found in educated self-KIR$^+$ NK cells. Blinded scoring of the size and staining intensity of 804 granzyme-B$^+$ structures (secretory lysosomes) from 20 single-KIR$^+$ NK cells, revealed that 50% of the educated NK cells carried at least one large secretory lysosome, representing on average 7.8% of the total number of secretory lysosomes per cell. Notably, in these cells, the large granzyme-B labeled structures contributed to 36% (8–76%) of the total granzyme B signal.

Optical resolution limits of confocal microscopy prevented accurate assessment of organelle size. To more precisely analyze the granzyme B distribution and lysosome morphology, sorted self-KIR$^+$ and non-self KIR$^+$ NK cells were analyzed by immuno-electron microscopy (immuno-EM) (Fig. 3e and Supplementary Figure 8). Quantification of gold particles per cellular section revealed overall greater granzyme B staining in self-KIR$^+$ NK cells (Fig. 3f), consistent with both the flow cytometry and confocal microscopy data. Since retention of granzyme B and re-loading of secretory lysosomes depend on the serglycin content in the matrix of the secretory lysosomes[27], sections of self-KIR$^+$ and non-self KIR$^+$ NK cells were stained for the expression of Chondroitin Sulphate 4 (CS4), a predominant glycosaminoglycan side-chain associated with serglycin in cytotoxic lymphocytes[28]. Self-KIR$^+$ NK cells had a higher overall intensity of CS4-staining

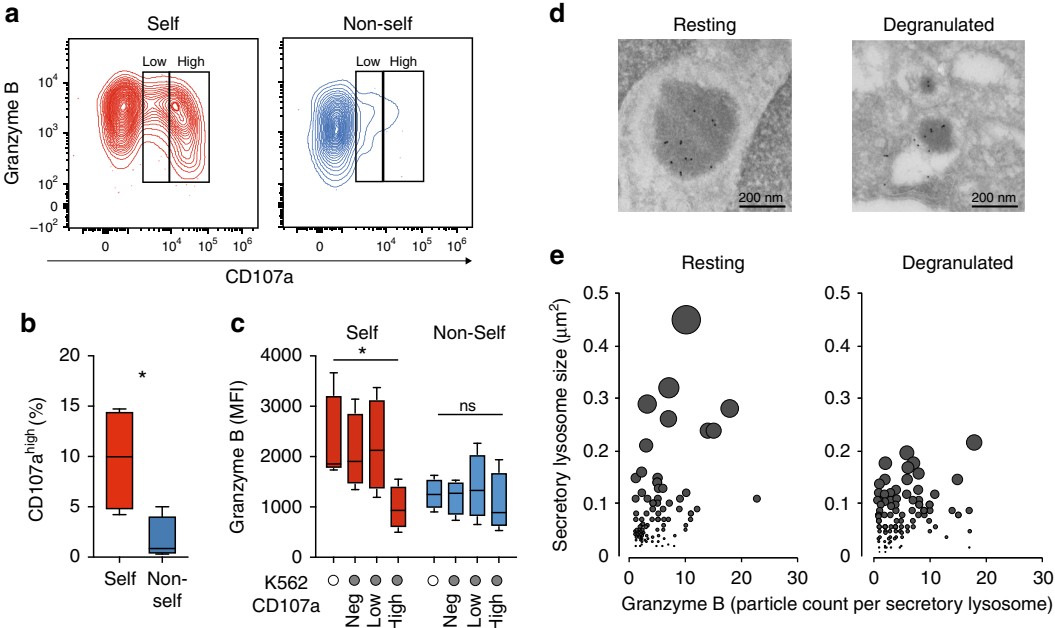

**Fig. 4** Educated NK cells mobilize dense-core secretory lysosomes. **a** Representative example of granzyme B and CD107a expression in self KIR[+] and non-self KIR[+] CD56[dim] NKG2A[−]NKG2C[−]CD57[−] NK cells following stimulation with K562 cells. **b** Aggregated data of percent CD107a[high] NK cells following stimulation with K562 cells ($n = 5$ donors). **c** Expression of granzyme B in the indicated NK cell subset after stimulation with K562. Summary of data from 4 C1/C1 donors. **d** Representative images of immuno-EM sections of resting (left) or sorted degranulated CD107a[high] NK cells (right). Scale bar: 200 nm. **e** Secretory lysosome size and granzyme B content as determined by immuno-EM in resting and sorted CD107a[high] NK cells after stimulation with K562. A Wilcoxon test was performed in panel (**b**). A non-parametric Friedman test was performed in panel (**c**). Whiskers show 5th to 95th percentile. Bars show the median. *$p < 0.05$. Red and blue box plots represent NK cells with self or non-self KIR, respectively

(Fig. 3e, f and Supplementary Figure 8), suggesting that NK cell education leads to changes in the matrix composition of the secretory lysosomes. Granzyme B and CS4 staining further revealed a small increase in the average secretory lysosome size, resulting in significantly larger total secretory lysosomal areas, without any difference in relative cell size (Supplementary Figure 7c–e). Similarly, the analysis of gold particle distribution of both granzyme B and CS4 against secretory lysosome area in immuno-EM images revealed that self-specific NK cells had larger granzyme B/CS4-dense lysosomal areas (Fig. 3g), and overall greater secretory lysosome densities (Fig. 3h). These data provide a link between the expression of self-specific inhibitory KIR and retention of enlarged, granzyme B-dense secretory lysosomes.

Given the accumulation of secretory lysosomes in educated NK cells, we examined the expression of other effector molecules, including perforin and granulysin in self and non-self KIR[+] NK cells. CD56[bright] NK cells lack secretory lysosomes and have low levels of both granzyme B and perforin (Supplementary Figure 9a). In contrast, granulysin was found at high levels also in CD56[bright] NK cells, suggesting that its production and storage are independent of the formation of dense-core secretory lysosomes (Supplementary Figure 9a). In support for a specific role of the increased density of secretory lysosomes in the retention of high levels of granzyme B in educated NK cells, perforin, but not granulysin was increased in self-KIR[+] NK cells (Supplementary Figure 9b–d). Importantly, the accumulation of effector molecules in educated self-KIR[+] NK cells was observed also in more differentiated NKG2A[−]CD57[+] NK cells (Supplementary Figure 9e).

**Educated NK cells mobilize dense-core secretory lysosomes**. It is well established that self-KIR[+] NK cells display stronger degranulation responses than non-self KIR[+] NK cells at the population level[2], which was corroborated in our functional

analysis of 96 donors (Fig. 1a). Furthermore, recent studies suggest that the release of as few as one secretory lysosome can lead to target cell killing[29]. Given the unique accumulation of granzyme B-dense secretory lysosomes in self-KIR[+] NK cells we examined their fate following stimulation with K562 cells. Granzyme B release in self-KIR[+] NK cells was associated with strong mobilization of secretory lysosomes, reflected in a higher discrete mode of CD107a expression (CD107a[high]) (Fig. 4a, b). High CD107a expression was accompanied by a decrease in granzyme B expression to the levels observed in non-self-specific KIR[+] NK cells (Fig. 4c). Immuno-EM of sorted CD107a[high] NK cells revealed a specific reduction of large and granzyme B-dense lysosomes following target cell interaction (Fig. 4d, e). Thus, the accumulation of dense-core secretory lysosomes during education and their effective release upon stimulation, provide a plausible explanation for the enhanced cytotoxic potential of self-KIR[+] NK cells and may contribute to their ability to perform serial killing[30].

**Compromising lysosomal activity decreases NK cell function**. NK cell education has a global influence on the function of self-KIR[+] NK cells extending beyond degranulation responses. These include increased Ca[2+] flux following receptor ligation, increased ability to form stable conjugates, and increased cytokine production following target cell interaction[13]. Therefore, we addressed whether there could be a link between the observed morphological changes in the lysosomal compartment and the known enhanced global responsiveness associated with NK cell education. There is increasing evidence suggesting that local Ca[2+] signaling from the acidic compartment, including secretory lysosomes, contributes to the spatiotemporal coordination of signaling cascades and boost Ca[2+] signaling from the endoplasmatic reticulum (ER)[31–34]. While most of the studies in this area have so far been performed in non-immune cells, there is evidence in both T cells and NK cells, that lysosomal Ca[2+] release

may play an important role in degranulation[31,35]. Therefore, we examined whether primary human NK cells were affected by the regulation of lysosomal activity and whether this had consequences on their global responsiveness to receptor ligation.

Functional responses of primary NK cells were determined in the presence and absence of glycyl-L-phenylalanine-beta-naphthylamide (GPN), a dipeptide substrate of cathepsin C associated with the release of $Ca^{2+}$ from the lysosomes[36]. GPN causes osmotic permeabilization of cathepsin C-positive compartments, resulting in the collapse of the pH gradient and controlled equilibration of small solutes, including $Ca^{2+}$, between the acidic compartment and the cytosol[36]. Treatment of resting primary NK cells with GPN dampened global $Ca^{2+}$-flux in the cytosol in response to ligation with CD16 or a combination of DNAM-1 and 2B4 (Fig. 5a). GPN treatment alone resulted in a low level of mobilization of CD107a[+] vesicles to the cell surface (Fig. 5b) but more importantly abrogated degranulation and the production of IFN-γ in response to K562 cells (Fig. 5b–d). Similar results were obtained using mefloquine, another lysosomotropic agent that

specifically disrupts lysosomal homeostasis through buffering of the acidic pH gradient (Fig. 5c, d)[37]. In line with the concept that NK cell education is not an on/off switch but rather a continuum of functional responses, both self KIR[+] and non-self KIR[+] NK cells were affected by lysosomal interference[38]. Importantly, none of these compounds showed any general cellular toxicity at the doses tested as compared to the positive control L-leucyl-L-leucine methyl ester (LeuLeuOMe), a lysosomotropic agent known to induce apoptosis in immune cells through induced lysis of secretory lysosomes (Supplementary Figure 10). Furthermore, GPN treatment did not interfere with degranulation in response to PMA/Ionomycin, which raises cytosolic-free $Ca^{2+}$ by directly accessing both intra- and extra-cellular-free $Ca^{2+}$ (Supplementary Figure 11). Thus, disruption of the lysosomal compartment not only affects the mobilization of secretory lysosomes but also the production of cytokines, suggesting that cytokine production in response to stimuli requires lysosomal-derived signals. This opens the possibility that differential

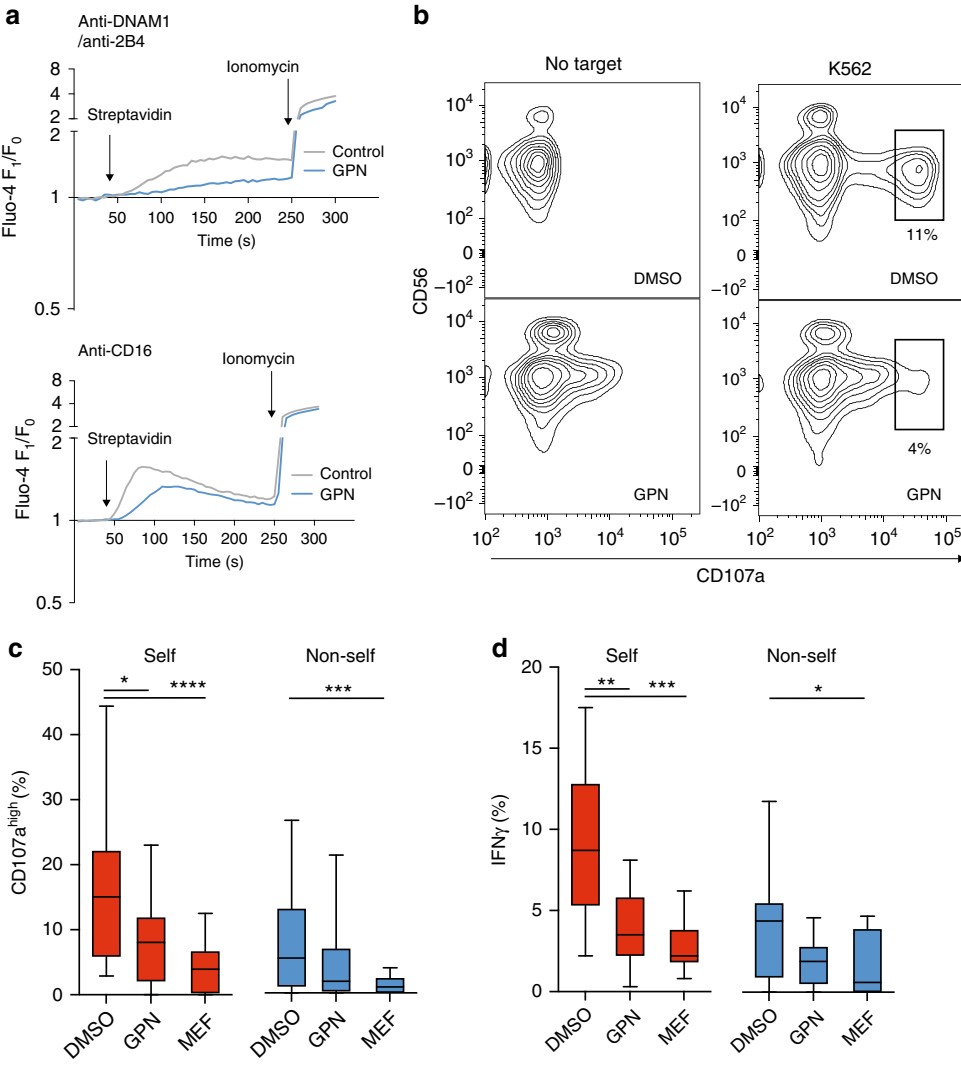

**Fig. 5** Compromising lysosomal activity decreases NK cell function. **a** Global $Ca^{2+}$-flux in resting bulk NK cells measured by Fluo-4 $F_1/F_0$ ratio following stimulation with biotinylated anti-DNAM-1/anti2B4 (top) or biotinylated anti-CD16 (bottom) crosslinked at the indicated time-point with streptavidin in the presence (added at the onset of stimulation and maintained throughout the assay) or absence of 50 μM GPN. **b** Representative example of CD107a expression following stimulation of NK cells with K562 cells in the presence or absence of GPN. **c** Frequency of CD107[high+] and **d** IFNγ[+] self-KIR[+] and non-self-KIR[+] NK cells following stimulation with K562 cells in the presence or absence of 50 μM GPN or 10 μM Mefloquine (MEF). Friedman's tests were performed followed by Dunn's multicomparison tests. Whiskers show 5th to 95th percentile. Bars show the median. ****$p < 0.0001$; ***$p < 0.001$; **$p < 0.01$; and *$p < 0.05$. Red and blue box plots represent NK cells with self or non-self KIR, respectively

signaling in self KIR$^+$ and non-self KIR$^+$ NK cells may be influenced by lysosomal-derived Ca$^{2+}$ signals.

**Enlarging secretory lysosomes increases NK cell function.** The lysosomal compartment undergoes constant modulation through Ca$^{2+}$ regulated fission and fusion events[39]. Lysosomal fission is dependent on Ca$^{2+}$ release via the lysosome-specific channel, transient receptor potential mucolipin-1 (TRPML1)[40,41].

TRPML1 is activated by phosphoinositide 3,5-bisphosphate PI (3,5)P$_2$[42], and may prevent uncontrolled fusion with secretory lysosomes[43]. To probe this axis, we blocked PI(3,5)P$_2$ synthesis in resting NK cells using three chemically distinct small molecule inhibitors of the PI3P 5 kinase, PIKfyve.

Treatment of cells with vacuolin-1, apilimod, and YM201636[44] led to the enlargement of the lysosomal compartment (Fig. 6a). Importantly, this was accompanied by a modest increase in

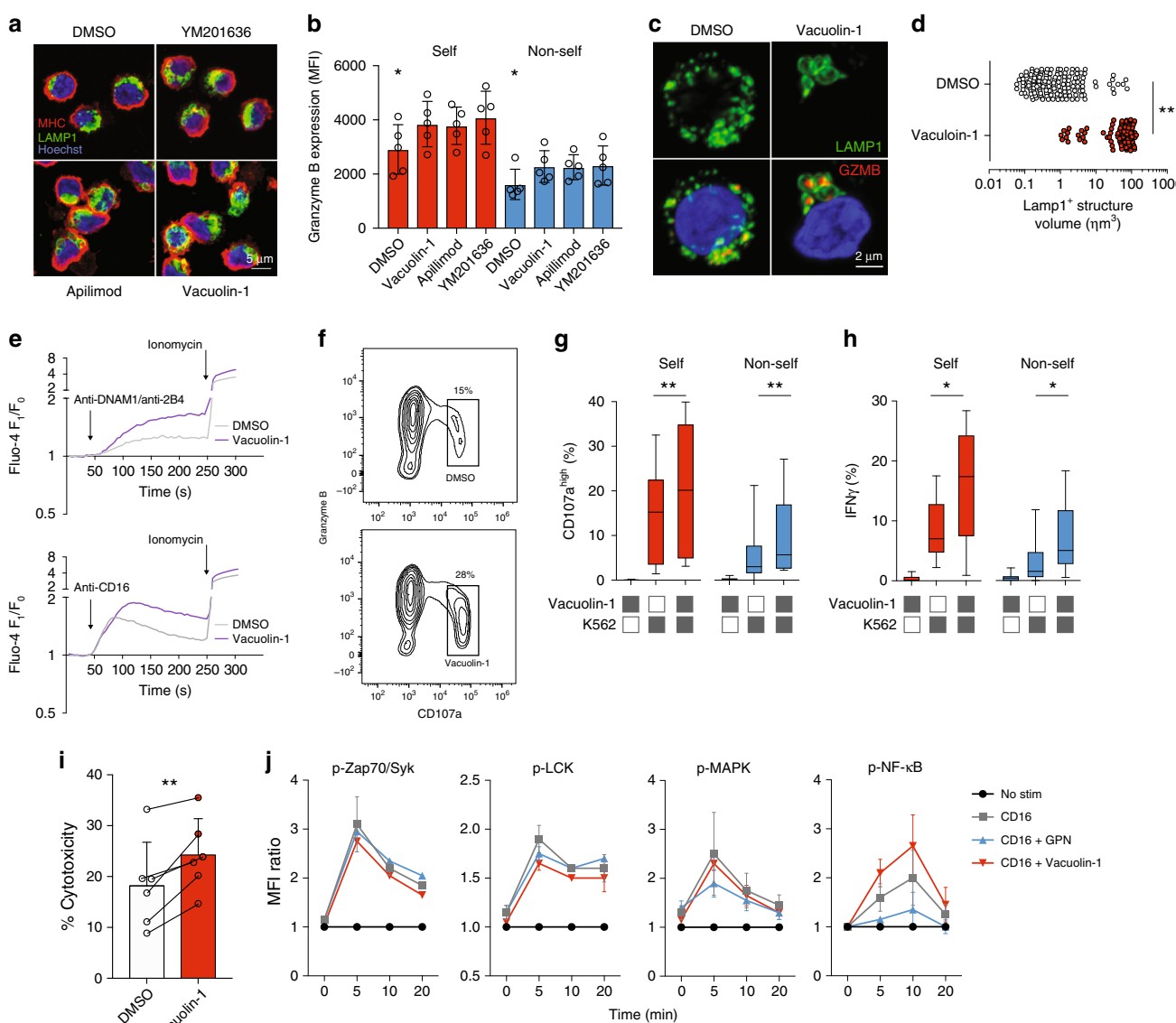

**Fig. 6** Enlarging the secretory lysosomes leads to enhanced NK cell function. **a** Confocal Z-stack showing MHC-I, LAMP-1, and granzyme B (GZMB) staining in primary NK cells following PIKfyve inhibition using overnight incubation with 1 μM vacuolin-1, 1 μM apilimod, or 1 μM YM201636. Scale bar is 5 μm. **b** Intracellular granzyme B expression in self-KIR$^+$ and non-self-KIR$^+$ NK cells following overnight incubation with the indicated PIKfyve inhibitor assessed by flow cytometry ($n = 5$ independent donors). **c** Representative example of a confocal image of primary NK cells treated overnight with 1 μM vacuolin-1 or DMSO. Scale bar is 2 μm. **d** Compiled confocal data on the volume of LAMP-1$^+$ structures from cells treated overnight with DMSO or vacuolin-1 ($n = 149$ LAMP-1$^+$ structures). **e** Cytosolic Ca$^{2+}$-flux in NK cells in response to stimulation with biotinylated anti-DNAM-/2B4 (top) or anti-CD16 (bottom) crosslinked with streptavidin at the indicated timepoint. Cells were treated with 10 μM vacuolin-1 added directly before the assay and then maintained throughout the incubation time. **f** Representative FACS histogram of granzyme B versus CD107a expression following stimulation with K562 cells in the presence of DMSO or 10 μM vacuolin-1. **g** Frequency of CD107a$^{high+}$ and **h** IFNγ$^+$ self-KIR$^+$ and non-self-KIR$^+$ NK cells after stimulation with K562 in the presence of DMSO or 10 μM vacuolin-1. **i** FACS-based killing assay showing NK cell killing of K562 cells after treatment with DMSO or 10 μM vacuolin-1 ($n = 6$ donors). **j** Relative phosphorylation of the indicated signaling molecules following stimulation with biotinylated anti-CD16 (10 μg/mL) crosslinked with streptavidin in the presence of 50 μM GPN or 10 μM vacuolin-1. Friedman's test was used in panel (**b**). A non-paired $t$-test was used in panel (**d**). Paired $t$-test was used in panels (**g–i**). Whiskers show 5th to 95th percentile. Bars show the median. ****$p < 0.0001$; **$p < 0.01$; and *$p < 0.05$. Red and blue circles and box plots represent NK cells with self or non-self KIR, respectively. In panels (**i**) and (**j**), red and blue colors indicate cells treated with the indicated compounds

granzyme B levels observed in both self-KIR[+] and non-self KIR[+] NK cells (Fig. 6b). Confocal microscopy of vacuolin-1-treated primary NK cells revealed localization of granzyme B within enlarged LAMP-1[+] structures (Fig. 6c, d). Treatment of NK cells with vacuolin-1 also increased global $Ca^{2+}$ flux in response to receptor-ligation (Fig. 6e) and enhanced specific degranulation (and mobilization of granzyme B) and IFNγ production in response to stimulation by K562 cells (Fig. 6f–h). Furthermore, the increased granzyme B expression and degranulation following PIKfyve inhibition correlated with increased natural cytotoxicity against K562 cells (Fig. 6i). These results demonstrate that chemical blockade of PIKfyve results in the enlargement of the lysosomal compartment and enhanced NK cell functionality.

In order to identify the point at which lysosomal disruption (GPN) or lysosomal enlargement (vacuolin-1) interfered with intracellular signaling pathways in NK cells, we probed signaling both proximal and distal to the plasma membrane. Vacuolin-1 had a minimal effect on upstream signaling, including ZAP70 and Lck following ligation of CD16 (Fig. 6j). However, the propagation of downstream signals through NF-κB was increased by treatment with vacuolin-1. Conversely, the disruption of lysosomal $Ca^{2+}$-flux by GPN had the reverse effect on CD16-induced NF-κB signaling (Fig. 6j). Hence, physical modulation of the acidic $Ca^{2+}$ stores affects downstream signaling in response to receptor ligation and tunes NK cell effector responses.

**TRPML1-mediated modulation of secretory lysosomes**. PIKfyve is recruited to PI3P positive compartments where it activates the lysosomal calcium channel TRPML1 via the production of PI(3,5)$P_2$[41]. The analysis of the transcriptional levels of TRPML1 in discrete NK cell subsets revealed TRPML1 mRNA was expressed at equal levels in all NK cell subsets (Fig. 7a). Agonistic stimulation of TRPML using the chemical compound MK6-83, which activates TRPML1 and TRPML3[45,46], resulted in the loss of granzyme B (Fig. 7b) and decreased specific degranulation and IFNγ responses to K562 cells (Fig. 7c). Conversely, silencing of TRPML1 by siRNA in resting primary NK cells (Fig. 7d) led to increased levels of granzyme B (Fig. 7e), confined within enlarged lysosomal structures (Fig. 7f, g). Moreover, in concordance with the effects of pharmacological inhibition of PIKfyve, siRNA silencing of TRPML1 led to enhanced degranulation and IFNγ production in primary resting NK cells (Fig. 7h–j). These results demonstrate a role for TRPML1 in the modulation of lysosomal structures, granzyme B content, and in tuning of effector function in NK cells.

## Discussion
NK cell education is a dynamic process during which NK cells calibrate their functional potential to self-MHC. However, it has been unclear how receptor input during NK cell education is integrated and retained in order for NK cells to remain self-tolerant, whilst also able to deliver spontaneous, well-tuned functional responses upon subsequent challenges. Our results suggest that unopposed activation signals lead to physical disarming of NK cells, mediated through TRPML1-induced modulation of the lysosomal compartment. The accumulation of dense-core secretory lysosomes under the influence of inhibitory self-MHC interactions provides mechanistic insights into the paradox of how inhibitory signaling is translated into a state of enhanced functional potential that persists between successive cell-to-cell contacts. The structural change in the lysosomal compartment and loading of dense-core secretory lysosomes may represent a form of molecular memory of receptor signaling during NK cell education.

A variety of models and nomenclature have been used to describe the process of NK cell education. However, regardless of whether the functional phenotype is caused by gain of function (arming/stimulatory licensing) in self-KIR[+] NK cells or loss of function (e.g., disarming/inhibitory licensing) in non-self KIR[+] NK cells[3,47], the net outcome of these processes is a consistent difference in the intrinsic functional potential of cells carrying self- and non-self receptors at rest. A structural basis for the difference in functional responsiveness has recently been proposed[9,10], whereby educated NK cells display unique compartmentalization of activating and inhibitory receptors at the nano-scale level on the plasma membrane. Complementing this pre-existing phenotype, we show that NK cell education is also tightly linked to the accumulation of large, granzyme B-rich secretory lysosomes, located closer to the centrosome in resting NK cells. We found a significant correlation between the expression of self KIR and the level of granzyme B for all three major inhibitory KIRs (2DL1, 2DL3, and 3DL1) and their cognate HLA ligands. Although the difference in granzyme B levels between educated and uneducated NK cells was only around 1.5–2-fold, the analysis of the size of the secretory lysosomes and total intracellular granzyme B expression before and after degranulation revealed a complete loss of large lysosomal structures and reduction of granzyme B down to baseline levels. These results suggest that the difference observed in flow cytometry is physiologically relevant and may constitute the whole releasable pool. Furthermore, these results are in line with the recent observation that NK cells can kill their target by releasing one single lytic granule[29]. It remains an open question whether the level of granzyme B stores is directly proportional to the functional capacity of the cell. We observed slightly higher levels of granzyme B in 3DL1sp NK cells from donors who possessed the Bw4[Ile80] allotype, which has been reported to be a high-affinity ligand[48], albeit this appears to depend on the KIR allele[49]. *3DL1* allelic diversity, including the non-expressed *3DL1*null alleles, combined with Bw4 ligand polymorphism has a profound influence on NK cell education[19,20,50]. We are currently addressing the impact of such allelic diversity on the granzyme B stores and graded functional responses. The observation that NK cell education is tightly linked to the cytotoxic payload has immediate implications for the cytotoxic potential of the cell. However, this finding alone cannot explain the extended functional phenotype of educated NK cells, namely their enhanced ability to form target cell conjugates and release IFNγ upon target cell stimulation[51]. Therefore, we set out to examine whether the remodeling of the lysosomal compartment could influence the functional potential in NK cell beyond the mere accumulation of effector molecules.

An increasing body of evidence supports the role of the acidic compartment not only in the triggering of $Ca^{2+}$ signaling, but also in the spatiotemporal coordination of signaling cascades[31–34], and the regulation of receptor degradation[52,53]. In both T cells and NK cells, lysosomal $Ca^{2+}$ release plays an important role in degranulation[31,35]. Signaling from the lysosomal compartment was also recently shown to regulate the migratory behavior of dendritic cells in a TRPML1-dependent fashion[54]. Our data suggest that there is a quantitative relationship between the modulation of the lysosomal compartment under the influence of inhibitory receptors and intrinsic functional potential of NK cells. While the exact role of the acidic $Ca^{2+}$ store for the enhanced functional potential in self-KIR[+] NK cells remains elusive, pharmacological inhibition of $Ca^{2+}$ release from the intracellular acidic stores, together with the analysis of $Ca^{2+}$-flux, consistently pointed to a role for the secretory lysosome in propagating surface receptor signaling. Notably, a correlation between the size of the lysosomes and level of $Ca^{2+}$-flux has previously been described in fibroblasts from patients with Parkinson disease[55].

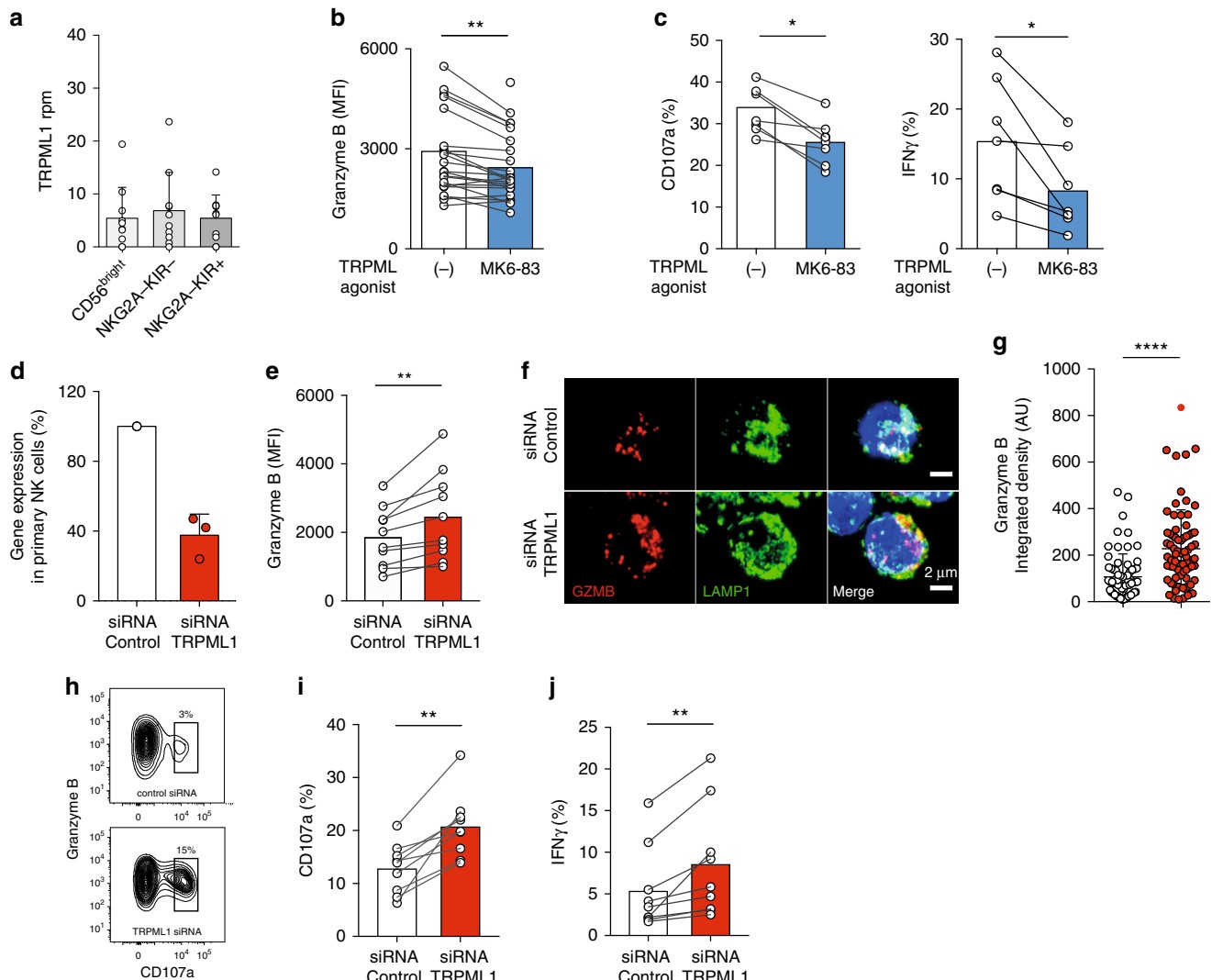

**Fig. 7** TRPML1-mediated modulation of secretory lysosomes. **a** mRNA expression (RNA-Seq) of TRPML1 in the indicated NK cell subsets sorted from PBMC and analyzed directly. **b** Granzyme B expression in NK cells treated for 2 h with 10 μM of the TRPML1 agonist MK6-83. (Summary of n = 23 donors.) **c** Degranulation (left) and IFNγ responses (right) by resting primary NK cells following stimulation with K562 cells for 4 h in the presence or absence of 10 μM MK6-83. Data are the summary from two independent experiments with 7 donors. **d** Relative mRNA expression (qPCR) of TRPML1 72 h after siRNA silencing in NK cells cultured for 3 days in 1 ng/mL IL-15. **e** Granzyme B expression in NK cells 72 h after siRNA silencing of TRPML1. **f** Confocal microscopy image showing LAMP-1 and granzyme B (GZMB) staining in siRNA TRPML1 silenced NK cells. Scale bar is 2 μm. **g** Summary of integrated granzyme B intensity per cell as quantified with ImageJ (n = 3 experiments). AU arbitrary units. **h** Representative example of FACS plot showing granzyme B expression versus CD107a in siRNA TRPML1 silenced NK cells after 4-h stimulation with K562 cells. Compiled data on **i** CD107a and **j** IFNγ production in TRPML1-silenced primary NK cells. Data are from nine donors with confirmed siRNA silencing. Paired *t*-tests were performed in panels (**b**, **c**, **e**, **i**, and **j**). Non-paired *t*-test was performed in panel (**g**). Whiskers show 5th to 95th percentile. Bars show the median. \*\*\*\**p* < 0.0001; \*\**p* < 0.01; and \**p* < 0.05. Red and blue box plots represent NK cells treated with the indicated compounds or siRNA

On that note, it is tempting to speculate that the gain of natural cytotoxicity in lL15-stimulated CD56^bright NK cells may be likewise related to the associated emergence of secretory lysosomes[56].

We examined the molecular pathway that led to the accumulation of secretory lysosomes in self-KIR+ NK cells, or rather the lack of accumulation of such lysosomes in non-self KIR+ NK cells. The difference in secretory lysosome size and densities in the absence of active lysosomal biogenesis led us to explore the pathways involved in the continuous modulation of the lysosomal compartment through fission and fusion events[39]. Several activating receptors have been implicated in NK cell education, including NKG2D, SLAM family receptors, and DNAM-1[7,11,57]. NKG2D and DNAM-1 signaling activates the PI3K/AKT

pathway[58]. Engagement of inhibitory receptors and SHP-1 signaling block NK cell activation at an early stage of the activation signaling pathway, preventing actin cytoskeletal rearrangement and the recruitment and phosphorylation of activation receptors[59]. The PI3K/AKT pathway is also controlled by SHIP1, which has also been implicated in tuning the effector function of NK cells during education[60]. The structural difference in the lysosomal compartment in NK cells lacking self-specific KIR led us to hypothesize that continuous unopposed signaling through activating receptors during homeostatic cell–cell interactions may promote lysosomal fission, leading to an inability to sequester granzyme B in large secretory lysosomes (model laid out in Fig. 8). Notably, PIKfyve and TRPML1 are activated downstream

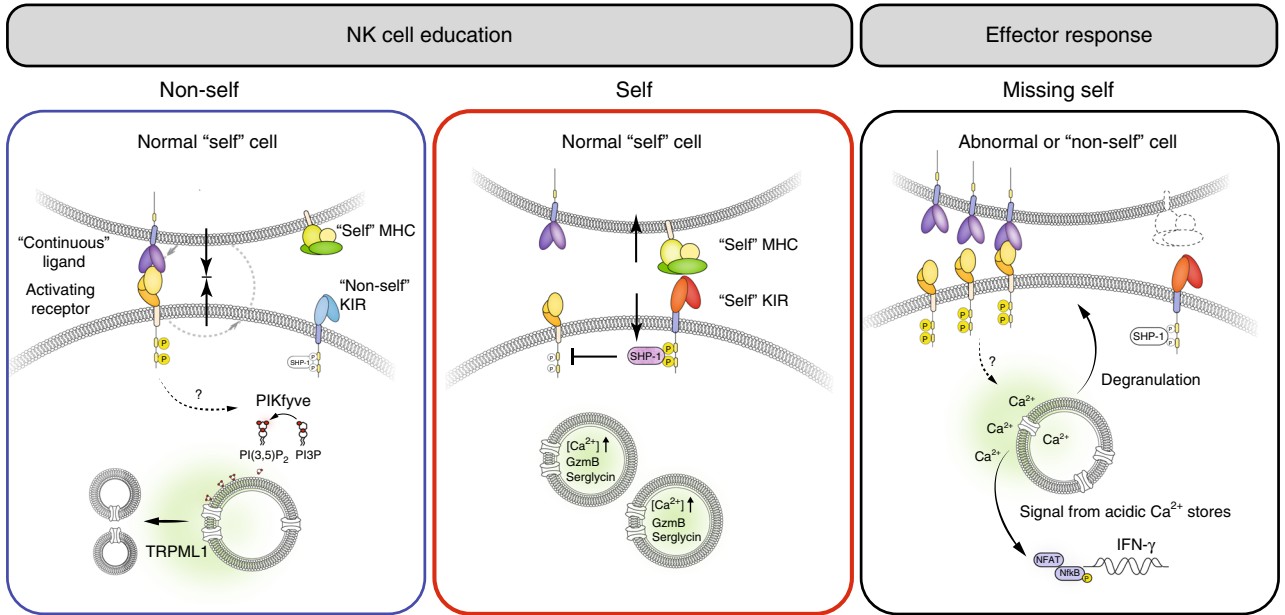

**Fig. 8** Model describing the distinct fates of NK cells during NK cell education. NK cells lacking self-specific receptors receive tonic stimulatory input through activating receptors and show poor functional responses, a process referred to as disarming[51]. We found that such cells exhibit lower levels of the granule matrix protein serglycin and effector molecules granzyme B and perforin and lack dense-core secretory lysosomes. One putative pathway downstream of activation receptor signaling is PI3K/AKT that stimulate the enzyme PIKfyve, which converts PI3P to PI(3,5)$P_2$ and thereby positively regulate the lysosome-specific $Ca^{2+}$ channel TRPML1[41]. PIKfyve and TRPML1 are critically involved in lysosomal modulation in several cell types[40, 41]. Inhibitory KIRs interfere with activation signals at a proximal level and thereby shut down any signals that could drive such lysosomal modulation. In support of this notion, we found that pharmacological interference with PIKfyve or silencing of TRPML1 replicated the educated state with enlarged lysosomes, increased granzyme B loads and more potent effector function. The secretory lysosome is part of the acidic $Ca^{2+}$ stores and may thus potentiate receptor-mediated $Ca^{2+}$ release from the ER[31-34]. Interference with signaling from the acidic $Ca^{2+}$ stores resulted in the loss of NK cell function. Thus, the accumulation of dense-core secretory lysosomes during NK cell education may contribute to the increased function, not only through the increased cytotoxic payload, but also through enhanced signaling from acidic $Ca^{2+}$ stores

of the PI3K/AKT pathway[52]. Mutations in TRPML1 cause mucolipidosis type IV, which is characterized by enlarged lysosomes[61]. The role of TRPML1 in lymphocytes is largely unknown. In other cell types, TRPML1 plays a role in lysosomal pH regulation[62], while more recent data support a role for TRPML1 in lysosomal fission[40]. Park et al. have suggested that TRPML1 guard against unintended, pathological fusion of lysosomes with other intracellular organelles, e.g., secretory vesicles[43]. TRPML1 has also been attributed to mediate lysosomal trafficking via $Ca^{2+}$-dependent motor protein recruitment, its activity favoring retrograde lysosomal movement[63].

To explore a possible role for PIKfyve and TRPML1 in lysosomal modulation in NK cells, we used a combination of pharmacological agonists and antagonists combined with genetic approaches to interfere with the PIKfyve/TRPML1 pathway. Pharmacological inhibition of PIKfyve by small chemical compounds, including vacuolin-1 and apilimod, is known to cause enlarged lysosomes in several cell types, including mast cells and macrophages[44,64,65]. Here, we show that the inhibition of PIKfyve by three different chemical compounds caused enlargement of the lysosomal compartment in NK cells. Importantly, this was associated with increased granzyme B expression, increased $Ca^{2+}$-flux, and more potent effector function, thus mimicking the educated NK cell state (Fig. 8). A similar functional phenotype was obtained when silencing the lysosome-specific $Ca^{2+}$ release channel TRPML1. Together, these data are compatible with a model where tonic or intermittent activation signals, possibly acting through the PI3K/AKT pathway, result in PIKfyve activation and TRPML1-induced lysosomal modulation, ultimately

leading to the lack of large secretory lysosomes and reduced functional potential in NK cells (Fig. 8).

In Chediak–Higashi Syndrome (CHS), mutation of the *LYST* gene leads to the formation of giant secretory lysosomal structures[66]. NK cells in CHS patients are hyperresponsive and hypersecretory but are unable to degranulate[67,68]. Although NK cell activation followed by secretory lysosome convergence and polarization appears to be normal in *LYST*-deficient NK cells, the enlarged secretory lysosomes fail to pass through the cortical actin meshwork openings at the immunological synapse[69]. By monitoring lysosomal size and granzyme B density prior to and following degranulation, we observed a selective loss of the pre-converged, large secretory lysosomes after degranulation. These large lysosomes had an area above 0.2 μm², corresponding to a diameter of around 500 nm, pointing to a possible difference in the density of the actin meshwork and physical restriction of degranulation between resting primary NK cells and NK92 cells. However, another contributing factor may be LYST-mediated modulation of lysosomal size during NK cell activation allowing smaller proportions of the large lysosomes to be released during the effector response, as has been demonstrated for degranulation in mast cells[70].

An outstanding question is how lysosomal fission leads to the loss of matrix component and lower levels of granzyme B in non-self KIR$^+$ NK cells. Granzyme B can be synthesized and secreted directly through the constitutive secretory pathway[71]. It is possible that an enhanced rate of lysosomal fission during weakly agonistic cell–cell interactions and the corresponding failure to accumulate dense-core lysosomes in NK cells lacking self-specific

receptors leads to the loss of granzyme through the secretory route[51]. Indeed, NK cells in serglycin$^{-/-}$ mice lack dense-core lysosomes, retain less granzyme B which is secreted from the cell at a greater rate, and exhibit reduced degranulation in response to stimuli[27].

Another remaining challenge is to decipher when and where TRPML1-mediated physical disarming takes place. Transfer experiments in mice have established an indisputable role for cell-to-cell interactions in shaping the functionality of mature NK cells[72,73]. Although the detailed time-scale and spatial aspects of such cell interactions remain largely unknown, transfer of functional NK cells to MHC-deficient environments leads to the induction of hyporesponsiveness[5]. SHP-1 intersects signaling of activating receptors upstream of Vav-1[59], and rapidly shuts down the process of forming an activating NK cell synapse with target cells[74]. While the inhibitory synapse and the productive cytolytic synapse have been studied in great detail, much less is known about immune synapses formed between resting immune cells during homeostasis. It is possible that cells lacking self-specific inhibitory receptors form a succession of non-cytolytic immune synapses under homeostasis leading to the loss of dense-core secretory lysosomes and leakage of their functional potential[51]. It has previously been shown that trans-presentation of IL-15 to NK cells, resulting in the activation of AKT is negatively regulated by inhibitory interactions with self MHC[75]. Thus, it is possible that unopposed constitutive IL-15 activation may occur in NK cells that lack self-specific inhibitory KIR, which in turn would affect lysosome stability and/or retention through the mechanism described here.

The dose-dependent induction of granzyme B expression in response to cytokine, or viral infection, is connected to the activation of the metabolic check-point kinase mTOR[76]. Notably, however, we did not observe any transcriptional imprint in the mTOR pathway when we examined circulating NK cells at rest, arguing against a major role for metabolism in the persistence of the distinct organization of the lysosomal compartment seen in circulating blood self-KIR$^+$ NK cells. It was recently shown that mTOR activation contributed to the functional rheostat during effector responses in educated murine NK cells[77]. Moreover, cytokine-activated educated human NK cells display unique metabolic regulation that influence their functionality[78]. Interestingly, mTOR activation and function are dependent on its lysosomal localization and the vacuolar H(+)-ATPase (V-ATPase) activity[79]. Furthermore, TRPML1 provides a negative feedback loop on mTOR activity[43]. Thus, the difference in lysosomal composition described in the present study could potentially contribute to enhanced mTOR activation and metabolic reprogramming observed in educated NK cells upon stimulation[77,78].

In conclusion, our findings suggest a mechanism by which NK cell education operates through modulation of the lysosomal compartment under the influence of inhibitory receptor–ligand interactions. Differences in the morphology of the lysosomal compartment and signaling from acidic $Ca^{2+}$ stores allow the cytolytic machinery to operate independently of transcription during the effector response. Furthermore, the data suggest that it may be possible to boost NK cell functionality through targeted manipulation of $Ca^{2+}$ homeostasis within lysosome-related organelles.

## Methods

**Cells**. Buffy coats from random healthy blood donors were obtained from the Karolinska University Hospital and Oslo University Hospital Blood banks with written informed consent. The approvals were obtained from the regional ethics committee in Stockholm 2006/229-31/3, 2016/1415-32, and the regional committees for medical and health research ethics in Norway: 2015/2095, 2015/2142, 2017/

420. Peripheral blood mononuclear cells were separated from buffy coats by density gravity centrifugation (Lymphoprep; Axis-Shield) using fretted spin tubes (Sep-Mate; Stemcell Technologies). Genomic DNA was isolated from 200 μl of whole blood using DNeasy Blood and Tissue Kit (Qiagen). *KIR* ligands were determined using the *KIR HLA* ligand kit (Olerup SSP) for detection of the HLA-Bw4, HLA-C1, and HLA-C2 motifs. NK cells were purified using negative selection (Miltenyi) with an AutoMACS Pro Separator. 221.Cw6 transfected with GFP were kindly provided by D. Davis, University of Manchester, England. K562 was purchased from ATCC. NKL and YTS cells were kindly provided by Dr. E. Alici, Karolinska Institute, Sweden. All cell lines were maintained in RPMI + 10% FCS and for NKL, the media was supplemented with 100 IU/mL of IL-2. All cell lines were maintained for a maximum of 20 passages and tested regularly for mycoplasma infection using the MycoAlert mycoplasma kit (Lonza).

**Phenotyping by flow cytometry**. Isolated PBMC were stained for flow cytometric analysis using an appropriate combination of antibodies as detailed in the Methods section. After surface staining, cells were fixed and permeabilized using a fixation/permeabilization kit (BD Bioscience Cytofix/Cytoperm) prior to intracellular staining with anti-granzyme B-A700 (GB11). Samples were acquired on LSRII or LSR Fortessa flow cytometers (both Becton Dickinson) and data was analyzed using FlowJo V10.0.8 (TreeStar). Fluorochrome-labeled antibodies used for phenotypic studies were as follows: CD14-V500 (M5E2), CD19-V500 or BV570 (HIB19), CD3-V500 or BV785 (UCHT1), CD56 ECD (N901), CD57-PB (HCD57), CD57-BV605 (QA17A04), CD57 purified (TB01), anti-mouse-IgM-EF650 (II/41), NKG2A-PE, APC or APC-AF750 (Z199), CD16-BV785 or BUV395 (3G8), KIR2DL3-FITC or Biotin (REA147), KIR2DL1-APC or APC-Vio770 (REA284), KIR3DL1-AF700 or BV421 (Dx9), KIR2DS4-QD585 (1847), KIR3DL2-biotin (Dx31), KIR2DL2/L3/S2-PE.Cy5.5 (GL183), KIR2DL1/S1-PE.Cy7 (EB6) or PE-Vio770 (11PB6). All mAbs were titrated and used at dilutions ensuring saturated staining of $1 \times 10^6$ cells. Dead cells were labeled using live/dead aqua (Life Technologies). Biotin-conjugated antibodies were visualized using streptavidin-Qdot 585 or 605 (Life Technologies) or BV711 (BD). Granzyme B-AF700 (GB11), Perforin-FITC (dG9), and Granulysin-AF488 (RB1).

**FACS sorting**. Purified NK cells were stained for FACS using the following combination: CD56-ECD, CD57-FITC, NKG2A-PE, KIR2DL1-APC-Vio770, KIR2DL1/S1-PE-Vio770, KIR3DL1/S1-APC (Z27.7.3), KIR2DL2/L3/S2-PE.Cy5.5 (GL183). Cells were sorted using a FACSAria at 4 °C (BD).

**RNA-Seq and qPCR**. RNA-Seq was performed using single-cell tagged reverse transcription (STRT), a highly multiplexed method for single-cell RNA-Seq. Real-time quantitative PCR was used to study the difference in the expression of genes of interest in sorted differentiation and education subsets of NK cells. Primer sequences are provided in Supplementary Methods.

**Confocal fluorescence microscopy and image analysis**. Sorted NK cells were prepared for confocal microscopy using fixation/permeabilization (BD Bioscience Cytofix/Cytoperm) prior to intracellular staining with mouse anti-human granzyme B-A647 (GB11), rabbit anti-human pericentrin (Ab4448) followed by Donkey anti-Rabbit IgG Alexa555. After staining, the fixed cells were adhered to glass coverslips using Cell Tak (Corning) and mounted using Pro-long Gold Antifade with DAPI. The cells were examined with a Zeiss LSM 710 confocal microscope (Carl Zeiss MicroImaging GmbH, Jena, Germany) equipped with an Ar-Laser Multiline (458/488/514 nm), a DPSS-561 10 (561 nm), a laser diode 405-30 CW (405 nm), and a HeNe-laser (633 nm). The objective used was a Zeiss plan-Apochromat 63× NA/1.4 oil DICII. Image processing and analysis were performed with basic software ZEN 2011 (Carl Zeiss MicroImaging GmbH, Jena, Germany) and Imaris 7.7.2 (Bitplane AG, Zürich, Switzerland). Confocal z-stacks were deconvolved using Huygens Essential 14.06 (Scientific Volume Imaging B.V., VB Hilversum, The Netherlands). ImarisCell was used to identify secretory lysosomes and centrosomes in confocal Z-stacks of single cells, while Imaris Venture was used to find the correlation between the intensity of the individual secretory lysosome and their distance to the centrosome center.

**Electron microscopy**. Sorted NK cells for immuno-EM were fixed in a mixture of 4% formaldehyde and 0.1% glutaraldehyde in 0.1 M PHEM buffer (60 mM PIPES, 25 mM HEPES, 10 mM EGTA, and 2 mM MgCl$_2$ at pH 6.9), followed by embedding in 10% gelatin, infiltration with 2.3 M sucrose and frozen in liquid nitrogen (LN$_2$). Ultrathin sections (70–90 nm) of cell pellets were cut on a Leica Ultracut (equipped with UFC cryochamber) at −110 °C, picked up with a 50:50 mixture of 2.3 M sucrose and 2% methyl cellulose. Sections were then labeled with antibodies against granzyme B (496B, eBioscience) or Chondroitin Sulphate 4 (2B6, AMSBIO), followed by a bridging rabbit-anti-mouse antibody (DAKO, Denmark) and protein A gold (University Medical Center, Utrecht, The Netherlands). Samples for Chondroitin Sulphate 4 staining were pretreated with chondroitinase ABC (AMS.E1028-02, AMSBIO) for 2 h at 37 °C according to manufacturer's recommendations. Microscopy was done at 80 kV in a JEOL_JEM1230 and images acquired with a Morada camera. Further image processing was done in Adobe

Photoshop. Quantification was done according to established stereological procedures.

**Functional assays**. Functional assays were performed at 37 °C in complete medium (RPMI + 10% FCS) for the times indicated. Purified NK cells were incubated with K562 target cells for 5 h at a ratio of 1:1 in the presence of anti-CD107a-alexa488 (H4A3, Biolegend) for degranulation assays, or with the addition of Brefeldin A (GolgiPlug BD) for degranulation plus intracellular cytokine assays. Lysosomotropic reagents were added immediately prior to the addition of targets/stimulation by agonistic antibodies and kept for the duration of the assay using the following final concentrations: GPN (50 µM), mefloquine (10 µM), and vacuolin-1 (1–10 µM). The TRPML1/3 agonist MK6-83 was used at 10 µM.

**Phospho-flow cytometry**. Functional assays for phospho-flow cytometry were performed at 37 °C in complete medium in NK cell suspensions between 5 and 10 M/mL for 20 min. Cells were pretreated for 1 h using GPN (50 µM) or Vacuolin-1 (10 µM), after which biotinylated CD16 (Biolegend, clone 3G8) was added to final concentrations of 5 µg/mL each. After 1 min, the aliquot for the 0 min (unstimulated) sample was taken out and mixed with Fix Buffer I (BD Biosciences). After 1 additional minute, the stimulation was started by crosslinking the biotinylated antibodies with 50 µg/mL avidin (Thermo Fisher Scientific) and the aliquots for the 5, 10, and 20 min samples were transferred into Fix Buffer I (BD Bioscience) at the corresponding time points. Cells were fixed at 37 °C for 10 min, washed and re-suspended in PBS. To allow combination of the differently stimulated samples into one, two-dimensional fluorescent cell barcoding (FCB) was utilized. Samples were stained in distinct concentrations of amine-reactive pacific blue succinimidyl ester (Thermo Fisher Scientific) for the time points (0 min—0.69 ng/mL, 5 min—6.25 ng/mL, 10 min—25 ng/mL, and 20 min—100 ng/mL) in combination with amine-reactive pacific orange succinimidyl ester (Thermo Fisher Scientific) for the different stimulations (control—10 ng/mL, GPN—100 ng/mL, and Vacuolin-1—500 ng/mL). After 20 min at RT, samples were washed twice in wash solution (PBS supplemented with 1% FCS and 0.09% sodium azide), combined, permeabilized (Perm Buffer III, BD Biosciences), and stored at −80 °C. For thawing, samples were incubated 20 min on ice. Then, they were washed in wash solution and stained with Alexa Fluor 647-conjugated phospho-epitope-specific antibodies against ZAP70/syk (pY319/pY352), Lck (pY505), Erk1/2 (pT202/pY204) (BD Bioscience), NF-κB p65 (pS536) (Cell Signaling Technologies) or isotype control IgG1κ (BD Biosciences) for 30 min at RT. After washing, data was acquired on an LSR Fortessa (BD Biosciences) and analyzed with FlowJo v10.0.8 (TreeStar).

**Ca²⁺ flux assay**. Freshly isolated NK cells were incubated with Fluo-4 for 30 min at 37 °C in PBS + 2% FCS at the recommended dilution (Fluo-4 Imaging kit, Molecular Probes). Cells were then washed twice and incubated with biotinylated CD16 or biotinylated DNAM-1/2B4 (Miltenyi), with the addition of labeled specific antibodies for CD56, CD57, NKG2A, KIR2DL1, KIR2DL1/S1, KIR3DL1/S1, and KIR2DL2/L3/S2 for 10 min at room temperature. The cells were washed once more and placed on ice until assayed. Prior to FACS analysis, the cells were pre-warmed at 37 °C for 5 min in the presence or absence of GPN (50 µM final concentration) or vacuolin-1 (10 µM final concentration). Cells were immediately run on FACS for 30 s, followed by the addition of 10 µg/mL streptavidin and run for a further 4 min. Ionomycin was added at 4 µM final concentration and run for a further 1 min. Ca²⁺-flux kinetics were analyzed by FlowJo V10.0.8 (TreeStar).

**Retroviral transduction of NK cell lines**. Full-length human *KIR2DL1* and *KIR2DL3* were synthesized using standard gene synthesis with codon optimization (Eurofins Genomics, Ebersberg, Germany), and subsequently subcloned into pMSCV using NotI and BamHI cloning sites. Retroviral particles were produced by transfection of Phoenix-ampho 293T cells using Lipofectimine 3000 (Life Technologies). YTS and NKL cells were spinoculated with viral supernatants for 60 min at 1000 × *g*. Cells were screened at five passages and positive cells were sorted using a FACSAria.

**siRNA interference**. Primary NK cells were isolated, rested for 2 h, and transfected either directly or primed with 1 ng/mL IL15 for 72 h and transfected. NK cells or cell lines were transfected by Amaxa nucleofection (Lonza) using 300 pM of Dharmacon ON-TARGET plus *SMART*pool control siRNA, or *SMART*pool RNA targeting human TRPML1. Nucleofection was performed using the human macrophage kit using program Y-010. After nucleofection, cells were rested for 4 h in OPTI-MEM, before an equal volume of culture medium with 2 ng/mL IL15 was added. The cells were then cultured for 48 h before phenotypic and functional testing. siRNA efficiency was determined using qPCR.

**Cytotoxicity assay by flow cytometry**. Target cell killing was determined using a combination of viability stains. Cytotoxicity assays were performed using an NK cell to target cell ratio of 5:1 at 37 °C for 5 h, after which cells were stained surface markers (CD56) to discriminate NK cells from target cells, with the addition of Live/dead aqua-fluorescent reactive dye (1:200; Life Technologies) for 20 min at 4 °C (or 15 min at RT), washed in staining buffer and stained in 50 µl RPMI media

plus 1 µM Yo-Pro®-3 iodide (Life Technologies) for 15′ at 37 °C. Finally, cells were washed and either directly analyzed using a BD™ LSR-II cytometer or fixed in 100 µl PFA 2–4% for 10 min at 4 °C, pelleted, washed twice with 200 µl staining buffer, and then rested at 4 °C until analyzed at the LSR-II machine.

**Incucyte**. YTS cells were incubated with GFP+ 221.Cw6 (C2) target cells at an E:T ratio of 1:1. The number of viable target cells was monitored by hourly fluorescence imaging over 24 h using an IncuCyte Live Cell Analysis System (Essen BioScience). The area of dead target cells, made up by GFP and Cytotox red (Essen BioScience) double positive cells, was quantified using IncuCyte Zoom software (Essen BioScience).

**RNA sequencing of education subsets**. RNA was isolated from sorted KIR single-positive NK cell subsets. Library preparation was performed using the Illumina NeoPrep Library preparation system. Sequencing was performed using the NextSeq (Illumina) (single read, 75 base pairs). Read alignment was carried out using Bowtie (version 2.0.5.0) and Tophat (version 2.0.6), and transcript abundance was estimated using Cufflinks (version 2.1.1). The resulting FPKM values for each transcript were log2 transformed for visualization in scatterplots (Fig. 2a).

**Quantitative PCR**. RNA was isolated using RNeasy mini kit (Qiagen). Following RNA isolation, cDNA was synthesized using First strand synthesis kit (Qiagen) according to manufacturer's protocol. Customized RT2 Profiler PCR array (Qiagen) was ordered with specific primers for the genes of interest, as well as two housekeeping genes, a reverse transcriptase control, genomic DNA control, and a positive PCR control (Supplementary Table 2). Real-time quantitative PCR was performed on cDNA from differentiation and education subsets and the data obtained was normalized using *18S rRNA* and *B2M* as housekeeping genes. All target genes were run as triplicates and analysis of qPCR data was done using qBase + (Biogazelle).

**Statistical analysis**. Statistical tests were selected based on the number of groups, pairing of samples, and whether or not the samples followed a Gaussian distribution and are indicated in the figure legends with reference to each panel. For comparison of two groups with a Gaussian distribution, paired or unpaired *t*-tests were used. ANOVA followed by multiple comparison tests were used for the analysis of more than two groups of samples with a Gaussian distribution. For comparison of two groups of unpaired samples with a non-Gaussian distribution, Mann–Whitney tests were used. For paired samples, a Wilcoxon test was selected. For comparison of more than two groups of paired samples with a non-Gaussian distribution, Friedman's test followed by Dunn's multiple comparison test was performed. For comparison of multiple unpaired groups with non-Gaussian distributions, Kruskal–Wallis test was used. n.s. indicates not significant; ****$p < 0.0001$; ***$p < 0.001$; **$p < 0.01$; and *$p < 0.05$. Analyses were performed using GraphPad Prism software.

## Data availability
All data, including images, generated and/or analyzed during the current study are available from the corresponding author on reasonable request. Due to the Norwegian data legislation, the RNA Seq datasets of anonymous donors are available upon request in the form of normalized FKPM values.

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

## Acknowledgements

This work was supported by grants from the Swedish Research Council, the Swedish Children's Cancer Society, the Swedish Cancer Society, the Tobias Foundation, the Karolinska Institutet, the Wenner-Gren Foundation, the Norwegian Cancer Society, the Norwegian Research Council, the South-Eastern Norway Regional Health Authority, and the KG Jebsen Center for Cancer Immunotherapy. S.P. was supported by BB/N01524X/1 from the BBSRC and B.J. was funded by a Mildred Scheel postdoctoral scholarship from the Dr. Mildred Scheel Foundation for Cancer Research of the German Cancer Aid Organization.

## Author contributions

J.P.G. designed and performed research, analyzed data, and wrote the paper. B.J., Q.H., D.C., E.S. and A.B. performed imaging or flow cytometry experiments and analyzed data. L.M.-Z., S. Lorenz, T.C., U.K., S. Linnarsson and M.L.S. performed RNA Seq and qPCR and analyzed data. W.E.L. contributed to the Ca$^{2+}$-signaling experiments. J.L. performed Phospho-flow experiments. A.P., M.T.W., E.H.A., L.L.L. and V.Y.S.O. performed experiments. S.P., C.G., K.T., and H.S. contributed to the design of research and the writing of the paper. K.-J.M. designed research, analyzed data, and wrote the paper.

## Additional information

**Competing interests:** K.J. Malmberg is a scientific advisor and consultant at Fate Therapeutics. J.P. Goodridge is currently employed as a Scientist at Fate Therapeutics (started after completion of this work). Malmberg and Goodridge are co-inventors on a patent application concerning the use of lysosomal modulation to tune NK and T cell function. The other authors declare no competing interests.

