## [Peer Review File · Nature Communications]

Reviewers' comments:

Reviewer #1 (NK development/function)(Remarks to the Author):

In the present study, Goodridge et al. show that educated NK cells expressing self-MHC specific inhibitory KIR (Self-KIR+) display accumulation of granzyme B in dense-core secretory lysosomes that converge close to the centrosome. This is suggested to be due to the TRPML1-induced modulation of the lysosomal compartment.

Despite new interesting technological approaches, there are major limitations that compromise the value of this study. In many cases, methodological details are not sufficient to allow replication of experiments. Legends to the figures are not sufficiently explanatory and lack analytical indications of the statistical tests performed. In some instances, there is not a correspondence between the main text and figures, both in terms of content and of referred panels. Concerning the secretory lysosomal content, it's quite surprising that only granzyme B (GzmB) has been evaluated while perforin has not even been mentioned. The study is composed of two main parts, the first is focused on Self-KIR+ versus Non-self-KIR+ NK cells (Figures 1-4), while the second on NK cells without such discrimination (Figure 5-7). This is clearly in contrast with the main topic of this article concerning NK cell education. Below there are some examples of criticisms/weaknesses mainly related to the first part of the Results that are crucial to support the novelty of this study.

- 1) A more detailed description of donors should be provided. In addition, donors should be analyzed for their KIR genotype. This is an important element to properly analyze data regarding the characterization and function (and thus education) of the NK cell subsets analyzed. For example, it is important to take into consideration the possible expression of KIR2DS2 and KIR2DL2, contributing to recognition of HLA-C1 alleles. More defined data for NK cell education would derive from donors having A/A KIR genotype. Since also GCN of KIRs modulates NK cell education, the analysis of A/A donors would avoid this further variable.
- 2) Gating strategies in the different experiments should be shown in Supplementary figures. It is not always clear whether single positive KIR co-express NKG2A. This is a crucial issue! In addition, has a cut-off number of events for each NK cell subsets been considered?
- 3) Figure 1a: the various CD107a percentages appear quite high, considering that resting NK cells were analyzed. It is not clear if these data represent degranulation upon K562 stimulation, like in Figure 4B, which shows lower but more realistic values.
- 4) Figure 1b: in Supplemental experimental procedure GzmB is not mentioned among the markers analyzed. In the legend, 10 donors are indicated, but in figure "donor" is indicated: is this a representative case? In the same donor two different populations should be compared: 2DL3 and 2DL1 with two different color borders both in C1/C1 and C2/C2 donors. The scale should be indicated. The exclusive expression of GzmB in Self-KIR+ cells is too impressive! Indeed, GzmB expression is not clearly pos/neg as for KIRs, but it's more a modulation of MFI bright/dim; moreover, also NKG2A+ cells should be GzmB+.
- 5) Figure 1d: the role of 3DL1 in education is rather complex, because it depends on both 3DL1 and Bw4 allele. An important reference regarding this issue is Saunders PM et al J Exp Med 2016.
- 6) Figure 1e: that a retroviral transduction to introduce full-length 2DL1 or 2DL3 into NK cell lines can reproduce a physiological education "in a dynamic model" appears forced. Moreover, there is no evidence that indeed the transduced KIR is functional.
- 7) Figure 2 and S2: phenotypic characterization (dot plots) of the different NK cell subsets before and after sorting should be shown. In global RNA-Seq, distribution of genes encoding for surface markers specific for each NK cell subset analyzed should also be shown. This is a control for the purity of sorted NK cell subsets analyzed.

Reviewer #2 (NK licensing, NKR)(Remarks to the Author):

In this manuscript, Goodridge and co-workers describe accumulation of granzyme B in secretory lysosomes, and a possible role of lysosome homeostasis in NK cell education. Education is a well-known outcome for NK cells co-expressing cognate inhibitory KIR and HLA partnerships but the mechanisms through which NK cell function is potentiated are largely undefined. The authors conclude that inhibitory KIR structural organization facilitates accumulation of granzyme B in dense-core secretory lysosomes (=lytic granules) in self KIR+ NK cells. The mechanism for these is interpreted to be a sort of "loading" of NK cells for rapid and efficient degranulation upon target cell recognition. That transfection of cell lines with self KIR could induce increased GrB expression is compelling toward their conclusion that inhibitory receptor expression is linked with NK cell education. The data are interesting, but some outstanding issues should be resolved and clarified before the conclusions of the manuscript can be drawn and the data can be interpreted as education/uneducation phenomena and not simply changes in lysosomal biology. As such, the authors have not been able to convincingly link lysosomal homeostasis to NK cell education as they have claimed in this manuscript.

Major concerns:

1. Figure 1. The authors use tSNE analysis to identify that KIR2DL1 cells in C2/C2 individuals and 2DL3 cells in C1/C1 individuals exhibit higher GrmB loading. How are the clusters defined? Additional tSNE plots showing 2DL1 and 2DL3 expression for both donors, as well as the localization of KIR3DL1+ cells are warranted. What is the HLA-B status/KIR3DL1 educating status in these donors? Are all donors haplotype-A/A? Otherwise, what is the GrmB loading among KIR2DL2 single+ NK cells? This would be especially interesting, given the relative promiscuity of KIR2DL2 binding to C1 and C2 (i.e. David J Immunol 2013; Frazier J Immunol 2013).
2. Given the variability in antibody clones and cross-reactivity especially, all of the specific clones used for staining human PBMC should be disclosed. What was the clone used for KIR2DL3, and did it cross-react with 2DL2 and 2DS1? Otherwise, how were these cells excluded from analysis? Likewise, how were KIR3DS1+ and KIR2DS1+ cells excluded? Given that activating receptors can be involved in NK cell education and cytotoxicity, these are important considerations. I recommend including a description of FACS analysis and a sample gating strategy be included.
3. Supplementary Figure 1a. Bw4+ KIR3DL1-low NK cells are still educated (albeit to a lower functional capacity than Bw4+3DL1-high, generally). The MFI of GrmB is NOT higher in this subpopulation, only in the high + Bw4 group. Was there further stratification of these donors based on the 80I vs. 80T alleles of Bw4, which are also known to modulate KIR3DL1+ NK cell education (i.e. Boudreau et al., JImmunol, 2016; O'Connor J Immunol 2014)?
4. In Figure 2b and Supplementary Figure 3, the authors assert that changes in transcription are separate from education, but do not actually consider donor KIR ligand background in these analysis. Separating the third group (NKG2a-KIR+) into NKG2a-selfKIR+ and NKG2a-non-selfKIR+ is required to draw any conclusion about education.
5. Figure S4a is meant to demonstrate greater granule size in self vs. non-self KIR+ cells. What are the red vs. white spaces? As is, it is not convincing to me that these are substantial differences in granule size. Likewise, Figure 3b attempts to quantify granule size, but does not include statistics. The majority of samples are clustered below 1×10^6 on the granule size scale, in both the self and non-self scales, so it is difficult to understand the authors' conclusions here.
6. For immuno-EM, figures are zoomed in to a single lysosome, but the authors indicate that a few

(average 3,2) large lysosomes are present in each NK cell, with far more smaller lysosomes. Figure 4d is therefore not convincing to me that there is a difference – these could just as easily be smaller lysosomes in an NK cell. The interpretation of this data would be assisted by photos of whole cells, where >1 “large” lysosome can be seen, especially to draw the conclusion that the large lysosomes shrink after cellular degranulation.

7. What is the consequence of degranulation of an uneducated cell? Degranulation of any cell under the presence of strong activating stimulation (i.e. antibody cross-linking or after stimulation with IL15 and IL21)? Is loss of GrmB/lysosome size associated with education or just depletion of GrmB stores in general? What is the total granule size in educated vs. uneducated cells? This conclusion may be better supported if some measure of variability in lysosome size were shown in aggregate data, comparing educated and uneducated NK cells from paired donors.

8. Watzl recently reported that GrmB is depleted after a few kills and NK cells switch to TRAIL and Fas-mediated killing. This is at odds with the authors’ interpretations that dense-core granule formations enable serial killing, but actually matches their data, that GrmB is “spent” in the most-degranulating (CD107a+) cells.

9. The interpretation that de-acidification of lysosomes prevents release of GrmB (figure 5) informs why educated NK cells have greater function is incorrect, and drawn on a correlation between the increased GrmB phenotype and educated cells. To draw this conclusion, the authors would need to compare educated and uneducated NK cells and show that educated NK cells were more impacted. The experiment, as is, simply demonstrates that impairing GrmB secretion impairs killing, but earlier in the paper, they demonstrate that GrmB can be upregulated in both educated and uneducated cells by IL-15 and IL-21 stimulation (Figure 2c). This implies that both populations of NK cells can kill via GrmB degranulation (and I would assume that both would be impaired by impairing that function).

10. As is, this paper demonstrates links from TRPML-1 to GrmB, lysosome formation and killing potential, but the link to KIR (and hence, education) is missing. The hypotheses laid out in the discussion should be tested to draw the link to education.

Response to reviewers' comments

Reviewer #1 (Remarks to the Author):

In the present study, Goodridge et al. show that educated NK cells expressing self-MHC specific inhibitory KIR (Self-KIR⁺) display accumulation of granzyme B in dense-core secretory lysosomes that converge close to the centrosome. This is suggested to be due to the TRPML1-induced modulation of the lysosomal compartment.

Despite new interesting technological approaches, there are major limitations that compromise the value of this study. In many cases, methodological details are not sufficient to allow replication of experiments. Legends to the figures are not sufficiently explanatory and lack analytical indications of the statistical tests performed. In some instances, there is not a correspondence between the main text and figures, both in terms of content and of referred panels.

Author response: In the revised manuscript, the legends have been improved and now include specific references to statistical tests performed in each panel. The materials and method section has been extended for further clarification of the experimental procedures. We have completed the editorial policy checklist for improved quality of methods and statistics that verifies compliance with all required editorial policies. See text marked in yellow throughout the manuscript.

Concerning the secretory lysosomal content, it's quite surprising that only granzyme B (GzmB) has been evaluated while perforin has not even been mentioned.

Author response: To address this point, we have extended the analysis of perforin and granzyme B in an extended cohort 49 of healthy donors (Data are shown in a new Supplementary Fig. 9, see below). Notably, as alluded to in the original submission, loading of positively charged granzyme B molecules has been shown to depend on the negatively charged glycoprotein serglycin, which is a key matrix component of the secretory lysosomes in cytotoxic lymphocytes. NK cells in serglycin KO mice lack dense core secretory lysosomes and display low levels of granzyme B (*ref 33: Sutton, V.R. et al. Serglycin determines secretory granule repertoire and regulates natural killer cell and cytotoxic T lymphocyte cytotoxicity. FEBS J 283, 947-961 (2016)*). Furthermore, NK cells in these mice are hypofunctional. In line with this observation, we found that educated NK cells expressed higher levels of chondroitin sulphate-4 (CS4), which is a key side chain of serglycin (**Fig. 3e-h and Supplementary Fig. S7c-d**). Although we could only verify this phenotype in immuno-EM due to lack of reagents for flow cytometry and confocal microscopy, the difference in CS4 expression, together with the difference in morphology (size and density) support the notion that education through self-specific KIRs is associated with structural changes in the lysosomal compartment, which in turn has consequences for the total granzyme B and perforin (**New Supplementary Fig. S9**) content in the cell. The structural changes in the lysosomal compartment hold important clues to why educated NK cells can carry more granzyme B and perforin in the absence of increased transcription or translation. The new data on perforin are discussed on page 11 with reference to Supplementary Fig. S9.

*Result page 11: "Given the accumulation of secretory lysosomes in educated NK cells, we examined the expression of other effector molecules, including perforin and granulysin in self and non-self KIR⁺ NK cells. CD56^{bright} NK cells lack secretory lysosomes and have low levels of both granzyme B and perforin (**Supplementary Fig. S9a**). In contrast, granulysin was found at high levels also in CD56^{bright} NK cells, suggesting that its production and storage is independent on the formation of dense-core secretory lysosomes (**Supplementary Fig. S9a**). In support for a specific role of the increased density of secretory lysosomes in the retention of high levels of granzyme B in educated NK cells, perforin, but not granulysin was found at higher levels in self-KIR⁺ NK cells (**Supplementary Fig. S9b**)."*

Figure S9 NK cell education through self-KIR is associated with accumulation of granzyme B and perforin but not granulysin. (a) Representative examples showing expression of granzyme B, perforin and granulysin in CD56bright and CD56dim NK cells. Expression of **(b)** granzyme B **(c)** perforin and **(d)** granulysin in NKG2A-CD57- NK cells expressing the indicated KIR in C1/C1 (n=16), C1/C2 (n=18) and C2/C2 (n=13) donors. KIR2DL1 single-positive (2DL1sp). Friedman's test followed by Dunn's multiple comparison test were performed in panels b-d. In panel d, there was no statistical accumulation of granulysin in any of the KIR+ subset.

The study is composed of two main parts, the first is focused on Self-KIR+ versus Non-self-KIR+ NK cells (Fig. 1-4), while the second on NK cells without such discrimination (Fig. 5-7). This is clearly in contrast with the main topic of this article concerning NK cell education.

Author response. We have stratified the outcomes of the pharmacological interventions based on expression of self-KIR and non-self KIR. As expected, the effect is more dramatic in educated NK cells that display a greater base-line function. However, manipulation of lysosomal function in uneducated NK cells led to weaker but corresponding differences (See revised Fig. 5 and 6). Overall, this aligns with the notion that education is a continuous rather than a discrete (On/Off) event where NK cells display graded responses based on the integrated input from the receptor repertoire.

Unfortunately, due to the need for high cell numbers, subset stratification was not possible to make in the gene silencing experiments shown in Fig. 7.

Below there are some examples of criticisms/weaknesses mainly related to the first part of the Results that are crucial to support the novelty of this study.

1) A more detailed description of donors should be provided. In addition, donors should be analyzed for their KIR genotype. This is an important element to properly analyze data regarding the characterization and function (and thus education) of the NK cell subsets analyzed. For example, it is important to take into consideration the possible expression of KIR2DS2 and KIR2DL2, contributing to recognition of HLA-C1 alleles. More defined data for NK cell education would derive from donors having A/A KIR genotype. Since also GCN of KIRs modulates NK cell education, the analysis of A/A donors would avoid this further variable.

Author response. This point is well taken. In the new extended cohort of 49 donors we have been able to stratify the granzyme B expression data in KIR subsets based on KIR haplotypes. The gating scheme and aggregated data are shown in new **Supplementary Fig. S1** (see below). KIR haplotypes per se did not influence the outcome. In haplotype B donors, we could resolve the levels of granzyme B in KIR2DL2 single positive NK cells revealing that it was similar in C1C1, C1/C2 and C2C2 donors (new Fig. 1e, see below). These results are discussed on page 7.

2) Gating strategies in the different experiments should be shown in Supplementary figures. It is not always clear whether single positive KIR co-express NKG2A. This is a crucial issue! In addition, has a cut-off number of events for each NK cell subsets been considered?

Author response. Agreed. See response to comment 1 above. We have included a general gating scheme in new **Supplementary Fig. S1a** with reference to the analysis of NK cell repertoires in Haplo A/A and B/x donors. NKG2A or CD57 co-expression is never permitted, unless those subsets were specifically analysed (for example in the analysis of NKG2A⁺ NK cells in Fig 1g). We have made that clearer in the revised legends. The general cut off we use is to quantify the MFI of at least 100 events in the final gate with only two exceptions down to >60 cells.

Figure S1. Gating scheme and analysis of granzyme B expression in haplotype A/A and haplotype B/x donors. (a) Gating scheme used to analyse expression of effector molecules in discrete NKG2A-CD57- NK cell subsets expressing single KIR. (b-c) Expression of granzyme B in NK cells triple negative (TN) for 2DL1, 2DL2/3, 3DL1 or expressing either of these KIR as their only KIR. Stratified analysis of donors with haplotype A/A and B/x in (b) HLA-C1/C1 and (c) HLA-C2/C2 donors. Red and blue colors indicate expression of self and non-self KIR, respectively. Friedman's test followed by Dunn's multiple comparison test were performed in panels b and c.

3) Fig. 1a: the various CD107a percentages appear quite high, considering that resting NK cells were analyzed. It is not clear if these data represent degranulation upon K562 stimulation, like in Fig. 4B, which shows lower but more realistic values.

Our experience is that the functional responses to K562 are very consistent within a given series of experiments but may vary significantly over longer time scales. All of these experiments in this manuscript were performed on resting fresh NK cells unless stated otherwise and the data were collected over 6 years using different instrumentation. The data are within the variation reported previously. We feel the key is to include all relevant controls in each experimental series to make side-by-side comparisons between subsets and/or treatments. We have improved the legends to help interpretation of the data in the different panels.

4) Fig. 1b: in Supplemental experimental procedure GzmB is not mentioned among the markers analyzed. In the legend, 10 donors are indicated, but in figure "donor" is indicated: is this a representative case? In the same donor two different populations should be compared: 2DL3 and 2DL1 with two different color borders both in C1/C1 and C2/C2 donors. The scale should be indicated. The exclusive expression of GzmB in Self-KIR+ cells is too impressive! Indeed, GzmB expression is not clearly pos/neg as for KIRs, but it's more a modulation of MFI bright/dim; moreover, also NKG2A+ cells should be GzmB+.

Author response. This comment is related to comment 1 by reviewer 2. We have revised the SNE plot to better represent all KIRs and the fine-tuned expression of granular content. We think it is a nice and complementary way to show the distribution of granzyme B across the repertoire but are aware that it does not provide additional information beyond what is shown in the classical graphs (Fig. 1c-d).

Updated technical description now moved into the method section (from supplement): FCS files from all donors, C1/C1 Bw4+ (n=4) and C2/C2 Bw4+ (n=2), were imported into FlowJo version 10.5.2 (TreeStar) and gated on CD14⁻ CD19⁻ CD3⁻ CD56^{dim} NKG2A⁻ CD57⁻ NKG2C⁻ live cells. These events were exported as FCS files for further processing using R version 3.5.1. 5500 events were randomly sampled from each file and the individual donors were then pooled for analysis. Arcsinh transformation with cofactor 150 was applied to all markers. Two-dimensional Barnes-Hut t-distributed SNE was performed with the Rtsne R package (<http://cran.r-project.org/package=Rtsne>) using standard settings. The SNE calculation was based on the markers KIR2DL1, KIR2DL3 and KIR3DL1. Plots were generated using the ggplot2 R package (<http://ggplot2.org>). For visualization, values below the 1st or above the 99th percentile were set to that of the 1st or 99th percentile, respectively. Blue borders indicating educated populations were added manually using Illustrator CS6 (Adobe).

Revised Fig. 1b. t-SNE plots showing intensity of granzyme B in clusters defined by KIR expression in C1/C1 Bw4+ (n=4) and C2/C2 Bw4+ (n=2) donors, respectively. The scale-bar is set for each marker and is therefore referred to as low to high relative expression.

5) Fig. 1d: the role of 3DL1 in education is rather complex, because it depends on both 3DL1 and Bw4 allele. An important reference regarding this issue is Saunders PM et al J Exp Med 2016.

See partly overlapping response to reviewer 2, point 3. We agree that analysis of 3DL1 is complicated by the effects of polymorphism in both 3DL1 and the Bw4 ligand. In the revised manuscript, we include a stratified analysis of Bw4 allotypes and discuss the data with reference to Saunders et al. (See new supplementary Fig. 2) and revised result section on page 7 and discussion on page 17.

See Discussion, page 17: “It remains an open question whether the level of granzyme B stores is directly proportional to the functional capacity of the cell. We observed slightly higher levels of granzyme B in 3DL1^{sp} NK cells from donors who possessed the Bw4^{lle80} allotype, which has been reported to be a high affinity ligand,¹¹ albeit this appear to depend on the KIR allele.¹² 3DL1 allelic diversity and Bw4 ligand polymorphism also influence NK cell education.^{13, 14} We are currently addressing the impact of such allelic diversity on the granzyme B stores and graded functional responses.”

6) Fig. 1e: that a retroviral transduction to introduce full-length 2DL1 or 2DL3 into NK cell lines can reproduce a physiological education “in a dynamic model” appears forced. Moreover, there is no evidence that indeed the transduced KIR is functional.

Author response. The natural variation in KIR gene expression within a given individual and across a cohort of donors provide a natural *in vivo* knock out model but still represent snap-shots of the status

at any given time. As pointed out by reviewer 2, the reciprocal data in the KIR-engineered NK lines provide additional support for a direct role of the self-specific KIR in generating the observed phenotype under conditions of NK cell priming.

7) Fig. 2 and S2: phenotypic characterization (dot plots) of the different NK cell subsets before and after sorting should be shown. In global RNA-Seq, distribution of genes encoding for surface markers specific for each NK cell subset analyzed should also be shown. This is a control for the purity of sorted NK cell subsets analyzed.

Author response. We have included a new supplementary figure showing representative plots of sorted self vs nonself KIR NK cell subsets (**Supplementary Fig. S4a**). We have also included data on RNA transcripts of KIRs, NKG2A and NKG2C in **Supplementary Fig. S4b**. CD57 is a carbohydrate epitope that is produced by the enzyme B3GAT1. Therefore, this gene is not informative for the purity of the sorts made. However, it is clear from the flow data that the cells analysed in downstream experiments were negative for CD57.

Reviewer #2 (Remarks to the Author):

In this manuscript, Goodridge and co-workers describe accumulation of granzyme B in secretory lysosomes, and a possible role of lysosome homeostasis in NK cell education. Education is a well-known outcome for NK cells co-expressing cognate inhibitory KIR and HLA partnerships but the mechanisms through which NK cell function is potentiated are largely undefined. The authors conclude that inhibitory KIR structural organization facilitates accumulation of granzyme B in dense-core secretory lysosomes (=lytic granules) in self KIR+ NK cells. The mechanism for these is interpreted to be a sort of “loading” of NK cells for rapid and efficient degranulation upon target cell recognition. That transfection of cell lines with self KIR could induce increased GrB expression is compelling toward their conclusion that inhibitory receptor expression is linked with NK cell education. The data are interesting, but some outstanding issues should be resolved and clarified before the conclusions

of the manuscript can be drawn and the data can be interpreted as education/uneducation phenomena and not simply changes in lysosomal biology. As such, the authors have not been able to convincingly link lysosomal homeostasis to NK cell education as they have claimed in this manuscript.

Major concerns:

1. Fig. 1. The authors use tSNE analysis to identify that KIR2DL1 cells in C2/C2 individuals and 2DL3 cells in C1/C1 individuals exhibit higher GrmB loading. How are the clusters defined? Additional tSNE plots showing 2DL1 and 2DL3 expression for both donors, as well as the localization of KIR3DL1+ cells are warranted. What is the HLA-B status/KIR3DL1 educating status in these donors? Are all donors haplotype-A/A? Otherwise, what is the GrmB loading among KIR2DL2 single+ NK cells? This would be especially interesting, given the relative promiscuity of KIR2DL2 binding to C1 and C2 (i.e. David J Immunol 2013; Frazier J Immunol 2013).

Author response. This comment is related to comment 4 by reviewer 1. We have revised the SNE plots using haplotype A/A donors to represent all KIR subsets in donors with relevant HLA backgrounds (see above). The SNE plots were generated by merging the phenotypes of 6 donors as described in the methods section (revised and moved from supplementary methods description). We also extended and stratified the data in Fig. 1c and confirmed the phenotypes in a set of 15 haplotype A donors (**Supplementary Fig. S1a-b**). In haplotype B/x donors, we analysed the expression of granzyme B in 2DL2 single positive NK cells. In keeping with the cross-reactive binding specific of 2DL2 to C1 and C2 the granzyme B levels were similar in C1/C1, C1/C2 and C2/C2 donors (**New panel: Fig. 1e**). This data is discussed on page 7 with reference to the papers above:

“Several alleles of KIR2DL2 (2DL2) have been shown to bind to both HLA-C1 and HLA-C2.^{9,10} In line with such cross-reactive binding patterns, 2DL2 single-positive NK cells expressed similar levels of granzyme B in HLA-C1/C1, -C1/C2 and -C2C2 donors (Fig. 1e).”

2. Given the variability in antibody clones and cross-reactivity especially, all of the specific clones used for staining human PBMC should be disclosed. What was the clone used for KIR2DL3, and did it cross-react with 2DL2 and 2DS1? Otherwise, how were these cells excluded from analysis? Likewise, how were KIR3DS1+ and KIR2DS1+ cells excluded? Given that activating receptors can be involved in NK cell education and cytotoxicity, these are important considerations. I recommend including a description of FACS analysis and a sample gating strategy be included.

Author response. Information of the antibody clones has been moved from the supplementary experimental procedures to the method section and the FACS analysis and gating strategies are shown in the new **Supplementary Fig. S1a**. As pointed out in the response to comment 1, the results were identical in haplotype A/A donors, lacking activating KIRs (Supplementary Fig. S1b). The only activating KIR with a clearly defined ligand that influence NK cell education is KIR2DS1 binding to HLA-C2. Through the gating scheme, we were able to analyse the expression of granzyme B in KIR2DS1 single-positive NK cells in a subset of Haplotype B/x donors. In line with the poor function of 2DS1 single positive NK cells in C2/C2 donors, the expression of granzyme B was low in this subset. These data are now shown in **Fig. 1f** (see below).

f

Revised Figure 1f. Expression of granzyme B in the indicated NKG2A⁺CD57⁻ NK cell subset in HLA-C2/C2 donors (n=7)

3. Supplementary Fig. 1a. Bw4+ KIR3DL1-low NK cells are still educated (albeit to a lower functional capacity than Bw4+3DL1-high, generally). The MFI of GrmB is NOT higher in this subpopulation, only in the high + Bw4 group. Was there further stratification of these donors based on the 80I vs. 80T alleles of Bw4, which are also known to modulate KIR3DL1+ NK cell education (i.e. Boudreau et al., J Immunol, 2016; O'Connor J Immunol 2014)?

Author response. This comment is related to comment 5 by reviewer 1. The functional diversity due to varying educating impact of 3DL1 alleles combined with variation in the Bw4 ligand makes this system very interesting but also harder to resolve without allele-level typing. This is partly the reason why we focused our initial efforts on the simpler dichotomy between C1 and C2 donors. In the revised manuscript, we include a stratified analysis of Bw4 allotypes. (**New Supplementary Fig. 2a-b, see below**) and have revised the result section on page 7 and discussion on page 17.

New Fig. S2. Granzyme accumulation through KIR3DL1 and cognate HLA-Bw4 ligands. (a) Expression of granzyme B in the indicated NKG2A⁻CD57⁻ NK cell subset from Bw4⁺ (n=26) and Bw4⁻ donors (n=16). TN=triple negative for 2DL1, 2DL3 and 3DL1. 3DL1sp=3DL1 single-positive. **(b)** Expression of granzyme B in 3DL1sp NK cells stratified based on the Bw4 ligands Bw4^{Thr80} (n=9), Bw4^{Iso80} (n=6), Bw4^{A24} (n=6). Other subtypes, eg., Bw4^{Thr80+A24} and Bw4^{Iso80+A24} were too rare to be analysed separately. One-way ANOVAs followed by Tukey's multiple comparison test were performed in panels a and b.

While the granzyme B phenotype clearly discriminated 3DL1 single positive NK cells educated by Bw4 versus those that were uneducated in Bw4 negative donors, the limited number of subjects precluded stratification based on allelic variants of 3DL1. We cite the above papers and discuss the question whether a graded granzyme B content could be a useful means to probe the educating impact of allelic variants when studied side-by-side with function.

See Discussion, page 17: *"It remains an open question whether the level of granzyme B stores is directly proportional to the functional capacity of the cell. We observed slightly higher levels of granzyme B in 3DL1sp NK cells from donors who possessed the Bw4^{Iso80} allotype, which has been reported to be a high affinity ligand,¹¹ albeit this appear to depend on the KIR allele.¹² 3DL1 allelic diversity and Bw4 ligand polymorphism also influence NK cell education.^{13, 14} We are currently addressing the impact of such allelic diversity on the granzyme B stores and graded functional responses."*

4. In Fig. 2b and Supplementary Fig. 3, the authors assert that changes in transcription are separate from education, but do not actually consider donor KIR ligand background in these analysis. Separating the third group (NKG2a-KIR⁺) into NKG2a-selfKIR⁺ and NKG2a-non-selfKIR⁺ is required to draw any conclusion about education.

Author response. Indeed, this is how these NK cells were sorted in **Fig. 2a** (for RNA Seq) and **Fig. 2b + Supplementary Fig. S6** (for confirmatory qPCR). We also controlled for differentiation by excluding CMV⁺ donors with large adaptive NK cell populations and by sorting CD57 negative NK cells. We have tried to state this more consistently throughout the text and figure legends. We also include a new **Supplementary Fig. S4a** to show pre and post sort phenotypes of self-KIR⁺ and non-self KIR⁺ NK cell subsets used for both RNA Seq and imaging. In the same new **Supplementary Fig. S4b** we also include a heatmap of mRNA reads (RNA Seq) on KIRs, NKG2A and NKG2C. The CD57 epitope is a carbohydrate epitope created by the enzyme B3GAT1, so it is not possible to verify its presence/absence by monitoring mRNA levels.

5. Fig. S4a is meant to demonstrate greater granule size in self vs. non-self KIR⁺ cells. What are the red vs. white spaces? As is, it is not convincing to me that these are substantial differences in granule size. Likewise, Fig. 3b attempts to quantify granule size, but does not include statistics. The majority of samples are clustered below 1x10⁶ on the granule size scale, in both the self and non-self scales, so it is difficult to understand the authors' conclusions here.

Author response. **Supplementary Fig. S7a** show representative examples of sorted NK cells that were used to quantify the distance of the granules from the centrosome. This was done blindly by an experienced microscopist at our imaging core (Ellen Skarpen). For increased clarity, we have revised the figure legend and included a color legend in the actual figure. Importantly, the stainings exemplified in **Fig. 3a** and **Supplementary Fig. S7a** were used for quantification of granzyme B pixel

intensity and distance from the centrosome as shown in **Fig. 3b (new), d** and **Supplementary Fig. S7b (new)**. These confocal images were NOT used for quantification of granule size. We have simplified **Fig. 3b** (by splitting into two parts) to allow statistical analysis of the results. These confocal imaging experiments allowed us to **i)** confirm the flow cytometry data in Fig. 1, showing greater intensity of granzyme B in self KIR⁺ educated NK cells (**Fig. 3b**) and **ii)** to calculate the distance from the centrosome, showing a unique accumulation of granzyme-B rich secretory lysosomes close to the centrosome (**Fig. 3d**). Visual inspection of the data in **Fig. 3d** indicated a general trend (in both subsets) towards localization of brighter secretory lysosomes closer to the centrosome. However, the average distance of the brightest secretory lysosomes (top 25%) was significantly shorter in self-KIR⁺ NK cells compared to non-self KIR⁺ NK cells (**Supplementary Fig. S7b**). The latter finding suggests that secretory lysosomes are converged in educated NK cells, even prior to activation. Given the limitation in resolution, granule size was examined by immuno-EM (see response to point 6 below). We hope the revised figures and figure legends help in clarifying this issue.

6. For immuno-EM, figures are zoomed in to a single lysosome, but the authors indicate that a few (average 3,2) large lysosomes are present in each NK cell, with far more smaller lysosomes. Fig. 4d is therefore not convincing to me that there is a difference – these could just as easily be smaller lysosomes in an NK cell. The interpretation of this data would be assisted by photos of whole cells, where >1 “large” lysosome can be seen, especially to draw the conclusion that the large lysosomes shrink after cellular degranulation.

Author response. We agree that showing representative images at lower magnification may help to visualize the data. New images of sorted educated and uneducated NK cells are shown in **Supplementary Fig. S8**. It is important to stress that not all granules were large in educated NK cells and that not all educated NK cells had large secretory lysosomes (these data are given in the text on page 10). This finding is not surprising or contradictory given that CD107a assays (the typical functional read-out used to probe education) also show heterogeneity in the sense that not all NK cells carrying a self-KIR are responsive. Unfortunately, there is no simple way to first observe the granular content by Immuno-EM and then probe the functionality of that same cell. By necessity this remains a correlation between high expression of granzyme B in NK cells that have self KIR and high frequency of responding NK cells among such cells.

Notably, we do not aim to infer that larger secretory lysosomes “shrink” after degranulation. We can only conclude that large granzyme-B dense structures are no longer found in any of the sections analysed. We cannot delineate whether this is due to release of one or two larger structures or through some other mechanism involving lysosomal modulation as described in mast cells (see discussion on page 21 and ref 86). Importantly, the notion that loss of few (one?) dense-core lysosomes in educated NK cells following degranulation (Immuno-EM data in **Figs. 3** and **4**) is associated with loss of granzyme B down to levels observed in uneducated NK cells (FACS data in **Fig. 4**) is compatible with the recent observation that one single granule is sufficient to kill a target cell (Discussed on page 17 with reference to the recent work of Jordan Orange).

7a. What is the consequence of degranulation of an uneducated cell?

The experiments presented in **Fig. 4a-c** examine the consequences of degranulation by resting educated and uneducated NK cells. We show, in agreement with the literature that the frequency of degranulating cells (eg CD107a⁺ cells) is low in uneducated NK cells. This is also shown in **Fig. 1a**. However, extending previous knowledge, our analysis shows that CD107a mobilization in uneducated NK cells does not lead to reduced levels of granzyme B, which reinforce the functional difference between the two subsets.

7b. *What is the consequence of* degranulation of any cell under the presence of strong activating stimulation (i.e. antibody cross-linking or after stimulation with IL15 and IL21)?

This is a very interesting question that clearly warrant further studies, although it is challenging to interpret the effects of cytokines when examining the upstream events leading to the educated state. Cytokine priming leads to increased mTOR activation and granzyme B levels (**Fig. 2c**) and typically to increased killing (previous extensive literature). As discussed in the original submission, even CD56^{bright} NK cells become granular, display increased loads of granzyme B, degranulate and kill following target cell stimulation (a study we recently co-authored Wagner et al., J. of Clinical Investigation 2018). While cytokine stimulation primes uneducated NK cells for increased function, our preliminary data show that there is still a slight difference in functionality between educated and uneducated NK cells following extensive cell division and phenotypic drift/or plasticity over the course of 6 days in IL-15 (*Pfefferle, manuscript in preparation. See Figure below for review only*).

We are currently exploring the detailed kinetics of these events, including the reloading of lysosomes and signals for lysosomal biogenesis driven by IL-15 and by the degranulation event itself. In contrast to process underlying NK cell education, such cytokine-driven responses are transcriptionally regulated and associated with subset plasticity as the cells are stimulated to undergo cell division. The proliferative response is tightly connected to differences in mTOR activation and metabolic differences as illustrated by a recent investigation in cytokine treated NK cells (Shafer et al. JACI 2018). To study cytokine-primed functional responses at the subset level is beyond the scope of the present study that focuses on the upstream cellular events leading to differences in lysosome morphology and granzyme B content. The data in **Fig. 2c-d** on cytokine responses are included only to rule out the possibility of an intrinsic difference in responsiveness to cytokines as a basis for their difference in granzyme B expression. We discuss the potential role of lysosomal signaling in the observed effects on mTOR and metabolism in relation to the novel study published by Lee et al. on page 22.

[Redacted]

7c. Is loss of GrmB/lysosome size associated with education or just depletion of GrmB stores in general?

As shown in **Fig. 4c**, the loss of granzyme B is unique to the educated NK cells, which is also the only subset that carry large granzyme-B dense secretory lysosomes. In this respect, our data support the

notion that loss of larger secretory lysosomes, and the correlated loss of granzyme B, is uniquely observed in educated NK cells. Concerning the granzyme stores and the refilling of those following degranulation and/or cytokine priming in both uneducated and educated NK cells, see response to point 6 and 7b above.

7d. What is the total granule size in educated vs. uneducated cells? This conclusion may be better supported if some measure of variability in lysosome size were shown in aggregate data, comparing educated and uneducated NK cells from paired donors.

The aggregated data on total granule area is shown **Supplementary Fig. S7d**. These data were quantified in a blinded fashion from sections of sorted educated and uneducated NK cells derived from 5 donors. Granule area per cell was determined by taking total calculated granule area per cell for structures staining positive for granzyme B. Unfortunately, not all comparisons were made pairwise which is why we used an unpaired non-parametric statistical test to evaluate the data. We think that the data describing the distribution of granzyme B-rich lysosomes (**Fig. 3g**) and the calculated density (**Fig. 3h**) is more relevant and therefore decided to show the total granular area as a supplemental figure.

8. Watzl recently reported that GrmB is depleted after a few kills and NK cells switch to TRAIL and Fas-mediated killing. This is at odds with the authors' interpretations that dense-core granule formations enable serial killing, but actually matches their data, that GzmB is "spent" in the most-degranulating (CD107a+) cells.

As far as we understand, this work is not yet published? We believe the preliminary data presented by C Watzl during the SNI2018 conference harmonizes well with our data, in particular the observation that granzyme B stores are never completely depleted even after serial killing. This aligns with the observation that educated NK cells lose granzyme B down to base-line levels observed in uneducated NK cells. This would indicate that uneducated NK cells do not participate in granzyme-B mediated serial killing. However, we agree that it is possible that uneducated may still participate in TRAIL/FAS-mediated (serial) killing. Therefore, we have softened the statement on the consequences of increase granzyme B on serial killing.

Page 12: "Thus, the accumulation of dense-core secretory lysosomes during education and their effective release upon stimulation, provide a plausible explanation for the enhanced cytotoxic potential of self-KIR⁺ NK cells and may contribute to their ability to perform serial killing.³⁶"

9. The interpretation that de-acidification of lysosomes prevents release of GrmB (Fig. 5) informs why educated NK cells have greater function is incorrect, and drawn on a correlation between the increased GrmB phenotype and educated cells. To draw this conclusion, the authors would need to compare educated and uneducated NK cells and show that educated NK cells were more impacted. The experiment, as is, simply demonstrates that impairing GrmB secretion impairs killing, but earlier in the paper, they demonstrate that GrmB can be upregulated in both educated and uneducated cells by IL-15 and IL-21 stimulation (Fig. 2c). This implies that both populations of NK cells can kill via GrmB degranulation (and I would assume that both would be impaired by impairing that function).

Author response. This is an interesting and somewhat challenging point to address. Our data show that interference of signaling from the acidic compartment with GPN and other lysosomotropic agents, has an impact on NK cell responses to receptor-ligation that extends beyond the release of the secretory lysosomes themselves. We also show that educated NK cells have a unique enlargement of the lysosomal compartment, suggesting that this may contribute to their enhanced functionality.

In the revised manuscript we have stratified the data based on expression of Self versus Non-Self KIR as suggested by the reviewer. These data show that the response is generally weaker in uneducated NK cells but as predicted this weak response is further diminished by interfering with lysosomal signaling (**new Fig. 5c-d see below**). Conversely, enlargement of lysosomal structures boost function in

uneducated NK cells albeit not to the levels observed in educated NK cells (**new Fig. 6g-h see below**). Hence, we agree with the reviewer that signaling from the acidic compartment is unlikely to be unique to educated NK cells. Our interpretation is that the magnitude of this response differs between self and non-self specific NK cells, owing to quantitative differences in the secretory lysosomal compartment observed in **Figs. 3-5**.

Revised Figure 5c-d. Compromising lysosomal activity decreases functional potential in NK cells. (c) Frequency of CD107a^{high} and **(d)** IFN- γ ⁺ NK cells following stimulation with K562 cells in the presence or absence of 50 μ M GPN or 10 μ M Mefloquine (n=32).

Revised Figure 6g-h. Enlarging the secretory lysosomes leads to enhanced NK cell functionality. (g) Frequency of CD107a^{high} and **(h)** IFN- γ ⁺ NK cells after stimulation with K562 in the presence of DMSO or 10 μ M vacuolin-1.

10. As is, this paper demonstrates links from TRPML-1 to GrmB, lysosome formation and killing potential, but the link to KIR (and hence, education) is missing. The hypotheses laid out in the discussion should be tested to draw the link to education.

In the revised manuscript we have tried our best to address all the specific concerns related to the data presentation and interpretation as well as performed new experiments. We hope that by revising the manuscript in accordance with the reviewer comments we have also built a more compelling case to support the revised model displayed in Fig. 8. We have made it more clear which parts of the cartoon are supported by the experimental data and which part of the model remains to be tested.

Revised Figure 8. Model describing the distinct fates of NK cells expressing self and non-self receptors during NK cell education and its consequence on effector responses. NK cells lacking self-specific receptors receive tonic stimulatory input through activating receptors and show poor functional responses, a process referred to as disarming.¹ We found that such cells exhibit lower levels of the granule matrix protein serglycin and effector molecules granzyme B and perforin and lack dense-core secretory lysosomes. One putative pathway downstream of activation receptor signaling is PI3K/AKT that stimulates the enzyme PIKfyve, which converts PI3P to PI(3,5)P₂ and thereby positively regulates the lysosome-specific Ca²⁺ channel TRPML1.^{2,3} PIKfyve and TRPML1 are critically involved in lysosomal modulation in several cell types.^{2,3,4} Inhibitory KIRs interfere with activation signals at a proximal level and thereby shut down any signals that could drive such lysosomal modulation. In support of this notion, we found that pharmacological interference with PIKfyve or silencing of TRPML1 replicated the educated state with enlarged lysosomes, increased granzyme B loads and more potent effector function. The secretory lysosome is part of the acidic Ca²⁺ stores and may thus potentiate receptor-mediated Ca²⁺ release from the ER.^{5,6,7,8} Interference with signaling from the acidic Ca²⁺ stores resulted in loss of NK cell function. Thus, the accumulation of dense-core secretory lysosomes during NK cell education may contribute to the increased function, not only through the increased cytotoxic payload, but also through enhanced signaling from acidic Ca²⁺ stores.

References

1. Goodridge, J.P., Onfelt, B. & Malmberg, K.J. Newtonian cell interactions shape natural killer cell education. *Immunol Rev* **267**, 197-213 (2015).
2. Pryor, P.R., Reimann, F., Gribble, F.M. & Luzio, J.P. Mucolipin-1 is a lysosomal membrane protein required for intracellular lactosylceramide traffic. *Traffic* **7**, 1388-1398 (2006).
3. Thompson, E.G., Schaheen, L., Dang, H. & Fares, H. Lysosomal trafficking functions of mucolipin-1 in murine macrophages. *BMC Cell Biol* **8**, 54 (2007).
4. Cao, Q., Yang, Y., Zhong, X.Z. & Dong, X.P. The lysosomal Ca²⁺ release channel TRPML1 regulates lysosome size by activating calmodulin. *J Biol Chem* **292**, 8424-8435 (2017).
5. Davis, L.C. *et al.* NAADP activates two-pore channels on T cell cytolytic granules to stimulate exocytosis and killing. *Curr Biol* **22**, 2331-2337 (2012).
6. Patel, S. & Docampo, R. Acidic calcium stores open for business: expanding the potential for intracellular Ca²⁺ signaling. *Trends Cell Biol* **20**, 277-286 (2010).
7. Patel, S. & Cai, X. Evolution of acidic Ca²⁺ stores and their resident Ca²⁺-permeable channels. *Cell Calcium* **57**, 222-230 (2015).

8. Wolf, I.M. *et al.* Frontrunners of T cell activation: Initial, localized Ca²⁺ signals mediated by NAADP and the type 1 ryanodine receptor. *Sci Signal* **8**, ra102 (2015).
9. David, G. *et al.* Large spectrum of HLA-C recognition by killer Ig-like receptor (KIR)2DL2 and KIR2DL3 and restricted C1 SPECIFICITY of KIR2DS2: dominant impact of KIR2DL2/KIR2DS2 on KIR2D NK cell repertoire formation. *J Immunol* **191**, 4778-4788 (2013).
10. Frazier, W.R., Steiner, N., Hou, L., Dakshanamurthy, S. & Hurley, C.K. Allelic variation in KIR2DL3 generates a KIR2DL2-like receptor with increased binding to its HLA-C ligand. *J Immunol* **190**, 6198-6208 (2013).
11. Cella, M., Longo, A., Ferrara, G.B., Strominger, J.L. & Colonna, M. NK3-specific natural killer cells are selectively inhibited by Bw4-positive HLA alleles with isoleucine 80. *J Exp Med* **180**, 1235-1242 (1994).
12. Saunders, P.M. *et al.* Killer cell immunoglobulin-like receptor 3DL1 polymorphism defines distinct hierarchies of HLA class I recognition. *J Exp Med* **213**, 791-807 (2016).
13. Boudreau, J.E., Mulrooney, T.J., Le Luduec, J.B., Barker, E. & Hsu, K.C. KIR3DL1 and HLA-B Density and Binding Calibrate NK Education and Response to HIV. *J Immunol* **196**, 3398-3410 (2016).
14. O'Connor, G.M. *et al.* Mutational and structural analysis of KIR3DL1 reveals a lineage-defining allotypic dimorphism that impacts both HLA and peptide sensitivity. *J Immunol* **192**, 2875-2884 (2014).

Reviewers' comments:

Reviewer #1 (Remarks to the Author):

The authors have improved the manuscript in the revised version by performing new experiments and including many changes based on the criticisms raised by the reviewers. However some criticisms persist.

Line 121: it is relevant to add the information that CD56dim are considered: "mature CD56dim NK cells".

From gating strategy (Fig. S1a) and M&M it appears that NKG2C+ cells are excluded, but this is not mentioned in the text, while NK cells in analyses are described as only NKG2A negative and CD57 negative (see line 136 and legend to Fig.1a). Please verify your data.

Fig. 1a: from the legend it does not appear that subsets are 2DL1 or 2DL3 single positive. Please specify because in the previous version they were indicated as single-positive NK cells.

Fig. 1b: data are now more believable, much less clear-cut than before. In my opinion, they could have chosen Bw4neg donors to avoid the involvement of 3DL1, which appears quite high and co-expressed with 2DL1 and 2DL3, and thus disturbing. Moreover Fig 1b is not completely according to what was said in line 143. Indeed, the level of GzmB in 3DL1+ 2DL1neg 2DL3neg cluster seems to be low, primarily in C2/C2 Bw4 donors, whereas the same cluster (educated by Bw4 both in the C1/C1 Bw4 and in the C2/C2 Bw4 donors) should express high levels of GzmB. Moreover, even in the revised version of t-SNE method, GzmB marker is still not mentioned. In my opinion, C2/C2 Bw4+ donors used to t-SNE analysis are few (only 2 donors) to be representative.

Line 147: to be corrected, 2DL2 single-positive should rather be termed 2DL2/S2 single-positive. In line 149 please substitute C2C2 with C2/C2.

Fig. 1e legend: in addition to 2DL2/S2 there are also 2DL3 single-positive.

Fig. 1h: in the Author response there is no mention that indeed the transduced KIR has an inhibitory function, as requested. Reverse ADCC data should be provided.

Fig. S4a: please indicate if the representative donor is C1/C1 or C2/C2. One can assume that donors are also having A/A KIR genotype, otherwise it is impossible to label cells as 2DL3+ and 2DL1+. The percentages of 2DL3sp and 2DL1sp appear similar, which is quite unexpected from a C1/C1 or C2/C2 donor. Also un-expected is the high amount of KIR- NKG2A- subset. In the legend, the inclusion of "CD56bright" to define the NK cell subsets appears a mistake.

Fig. 5b: "a representative example of granzyme B and CD107a" is written in the legend, however CD56 and CD107a are shown.

Fig. 5c,d: no comment is provided in the text mentioning that self versus non-self NK cells were analyzed.

Reviewer #2 (Remarks to the Author):

Although I remain optimistic that the authors' observations will be informative toward the perplexing question of NK cell education, I remain unconvinced that the pathway has been conclusively determined in this study. I agree that calcium flux, GrB accumulation, and lysosome size and localization are all correlates to NK cell effector potential, the data are insufficient to link education to managing these pathways. For instance, there is no functional data to support a link between KIR/SHP-1/ITIM (which are known to be critical for education) in manipulating the lysosomes, and this would be needed to draw the conclusion that this is a phenomenon rooted in education. The data are sufficient to conclude that effector potential is linked to the phenotypes they describe, but the finding that cytokine stimulation can phenocopy these outcomes suggests that they are instead associated

with NK cell effector potential (which is known to be higher at baseline and may be controlled by a third party pathway not identified).

1. Figure S2 shows KIR3DL1 single + cells stratified by Bw4 status (A) and Bw4 subtype (B). The number of Bw4+ cells in A differs from those shown in B (presumably, the Bw4+ bar in A is simply a further stratified version of A). Why were 5 donors excluded from panel B? Even if these events were among the A24+HLA-Bx which were “too rare to be analyzed” they should be shown on panel B. Moreover, Boudreau et al., J ClinOncol 2017 demonstrated that HLA-B alleles’ impact on NK cell education superceded HLA-A*24 alleles, providing precedence that these should be included in the Iso or Thr groups (and A24 should only be composed of those with Bw6/Bw6).

2. The authors have declined the request to stratify data by KIR3DL1 subtype due to low cell numbers. Boudreau et al (JI 2016) show that density predicts NK cell education based on its density. Perhaps an alternative approach to allele typing would be to correlate KIR3DL1 MFI with GrB density and/or to gate on the particular subpopulations of KIR3DL1 on FACS plots, where the populations can be clearly distinguished. Demonstrating a dose-response correlation would be very powerful here.

3. Boudreau et al (J ClinOncol 2017) showed that the null subtype of KIR3DL1 could drive NK cell education. How were KIR3DL1-null NK cells dealt with? NK cells from donors exhibiting this allele may co-express KIR3DL1-n and therefore not be gateable.

4. Response to reviewer 2, 7b: While I realize that the study of cytokine-driven NK cell degranulation differs from that triggered by a missing self reaction, it is NOT beyond the scope of this paper. The authors aim to make conclusions about the differences of NK cells based on NK education. This can be accomplished in at least 2 ways: first, by comparing so uneducated NK cells from the same donors, which I acknowledge that the authors have done, and second, by proving that this has anything to do with education. I acknowledge that the responses may be different and that is the point of the query: Is the phenotype associated with education, or simply with NK cell potentiation (which would be more broadly defined)?

5. If the authors aim to show that self KIR+ NK cells are more potentiated (and more impacted by lysosomal manipulations) in figures 5 and 6, they should compare (pairwise by donor) based on treatment group between s and ns KIR.

6. New figure 8 is a speculation of the model that links their data to education. I do not oppose the inclusion of a model system, but the link between putative activation (the disarming model has NOT itself been conclusively proven) and AKT activation is testable and, this paper’s data alone cannot draw the vector from activation to PIKfyve activation. The statement that the authors have indicated where data are speculative is not sufficient to mitigate this problem (especially since it is written in a caption that requires further activation). At a MINIMUM, speculation should be indicated with dashed lines and question marks in the speculative model.

7. The MFI of GrB in figure 6 implies that the cells have been selected for self-KIR+, but it is not explicitly stated that this is the case. Does inhibition of PIKfyve lead to the same phenotype in educated and uneducated cells? If it does, this may assist in developing the mechanistic conclusion that lysosome size and GrB loading is a direct correlate to NK cell potentiation.

8. If paired t-tests are used in Figure 7, the authors should also include some indicator of which samples are the pairs (either by specific colors or joining lines). As presented, the data are unconvincing of differences. I note that there are different numbers of samples in the panels and throughout the manuscript. Were power calculations undertaken to determine the optimal number of

replicates?

9. If activating stimulation is required, what is its source in the transfection models shown in Figure 1h? That cells exhibit an education-like phenotype with respect to GrB after this transfection supports education by inhibitory interactions with self MHC.

10. The color bar scale is missing from Figure 1b in the manuscript (but not rebuttal).

11. Separation of KIR2DL2 single positive cells is still unclear to me. The combination of antibodies for KIR2DL3 and KIR2DL2/L3/S2 enables isolation of KIR2DL3 from this group, but how is KIR2DS2 excluded?

12. Many educated NK cells exhibit CD57. Why were cells exhibiting this marker excluded? At a minimum, I recommend including a description of the data for CD57+ NK cells.

13. In figures 2a and b, the authors sort KIR+ NK cells (but not self-KIR+) and draw the conclusion that self-KIR+ NK cells develop according to the transcriptional program they describe. This conclusion cannot be made without comparing to non-self KIR+ NK Cells AND separating the self from the non-self populations.

Reviewers' comments:

Reviewer #1 (Remarks to the Author):

The authors have improved the manuscript in the revised version by performing new experiments and including many changes based on the criticisms raised by the reviewers. However some criticisms persist.

Author response: We thank the reviewer for this positive remark and for carefully scrutinizing the revised manuscript.

Line 121: it is relevant to add the information that CD56dim are considered: “mature CD56dim NK cells”.

Author response: we have added “mature” on page 8, line 15 and in the figure legend of Figure 1 and 3.

From gating strategy (Fig. S1a) and M&M it appears that NKG2C⁺ cells are excluded, but this is not mentioned in the text, while NK cells in analyses are described as only NKG2A negative and CD57 negative (see line 136 and legend to Fig.1a). Please verify your data.

Author response: We have excluded donors with adaptive expansions with high frequencies of NKG2C and gated on NKG2C negative NK cells. Adaptive NK cells are transcriptionally and epigenetically different and express high levels of granzyme B which is why it is important to make this distinction. This information is now added in the legend in Fig.1a and in the text where relevant.

Fig. 1a: from the legend it does not appear that subsets are 2DL1 or 2DL3 single positive. Please specify because in the previous version they were indicated as single-positive NK cells.

Author response: The subsets are indeed single-positive for self and non-self KIR, respectively. This information is now added in the legend. We apologize for this unintentional omission and the confusion caused.

Fig. 1b: data are now more believable, much less clear-cut than before. In my opinion, they could have chosen Bw4neg donors to avoid the involvement of 3DL1, which appears quite high and co-expressed with 2DL1 and 2DL3, and thus disturbing. Moreover Fig 1b is not completely according to what was said in line 143. Indeed, the level of GzmB in 3DL1⁺ 2DL1neg 2DL3neg cluster seems to be low,

primarily in C2/C2 Bw4 donors, whereas the same cluster (educated by Bw4 both in the C1/C1 Bw4 and in the C2/C2 Bw4 donors) should express high levels of GzmB. Moreover, even in the revised version of t-SNE method, GzmB marker is still not mentioned. In my opinion, C2/C2 Bw4+ donors used to t-SNE analysis are few (only 2 donors) to be representative.

Author response: We have deleted the SNE plot since it was mostly confusing and provided no additional information compared to the robust flow cytometry data shown in several panels in Figure 1.

Line 147: to be corrected, 2DL2 single-positive should rather be termed 2DL2/S2 single-positive.

Author response: Indeed. We have corrected the figure and legend, as well as the text.

In line 149 please substitute C2C2 with C2/C2.

Thanks, corrected.

Fig. 1e legend: in addition to 2DL2/S2 there are also 2DL3 single-positive.

Thanks for noticing, corrected.

Fig. 1h: in the Author response there is no mention that indeed the transduced KIR has an inhibitory function, as requested. Reverse ADCC data should be provided.

Author response: Unfortunately, the KIR engineered lines show considerable drift with loss of the transgene and we are currently remaking all of these lines from scratch. It is taking longer than anticipated to remake these lines and we do not know when we can provide robust data on the inhibitory function of the KIR transduced lines. Others have shown inhibitory function of KIR engineered NK92 and YTS cells, so it is well documented that these NK lines are receptive to inhibitory signaling.¹ However, to address this concern in reasonable time, we decided to include one representative long-term killing experiment previously performed using the Incucyte imaging platform. In these experiments we tested YTS (C1/C1) transduced with 2DL1 or 2DL3 against 221.Cw6 (C2). This experiment corroborates the observation of an educating impact mediated by the self KIR (2DL3) and show inhibition from baseline (YTSnil) in YTS.2DL1 cells interacting with 221.Cw6 targets. Unfortunately, we do not have the corresponding graphs for 221.wt targets that was not included in this experiment. This figure is included in Supplementary Figure S3 (see below). The new Incucyte experiment is described in the method section.

Figure S3. Functional responses in KIR-transduced NK cell lines. Long-term cytotoxicity assay in the Incucyte showing killing (GFP+Cytotox Red+ area) of 221.Cw6.GFP+ cells by the KIR engineered YTS lines E:T ratio 1:1. Bars represent SD of triplicates.

Fig. S4a: please indicate if the representative donor is C1/C1 or C2/C2. One can assume that donors are also having A/A KIR genotype, otherwise it is impossible to label cells as 2DL3+ and 2DL1+. The percentages of 2DL3sp and 2DL1sp appear similar, which is quite unexpected from a C1/C1 or C2/C2 donor. Also un-expected is the high amount of KIR- NKG2A- subset. In the legend, the inclusion of “CD56bright” to define the NK cell subsets appears a mistake.

Author response. Indeed, these sorts were performed in haplotype A/A donors which is also reflected in the RNA Seq data. The “CD56^{bright}” was indeed a typo and has been deleted. The selected donor was C1/C1. The legend has been updated in the revised manuscript. Our laboratory has studied repertoire formation and diversity in detail in several papers.^{2,3,4,5} Although 2DL3 frequencies are normally higher than 2DL1 frequencies (regardless of C1/C1, C2/C2 status), the frequency of 2DL3 and 2DL1 varies significantly between donors and this particular donor is within the normal variation. The same is true for NKG2A-KIR- NK cells that vary between 4 and 44%.⁶ For these experiments it was essential to have significant proportion of both subsets for downstream analysis.

Fig. 5b: “a representative example of granzyme B and CD107a” is written in the legend, however CD56 and CD107a are shown.

Author response. Thanks for noticing this typo that has now been corrected.

Fig. 5c,d: no comment is provided in the text mentioning that self versus non-self NK cells were analyzed.

Author response. Thanks for noticing this omission. We have added this information in the text and in the legends in Figure 5 but also in Figure 6.

Reviewer #2 (Remarks to the Author):

Although I remain optimistic that the authors’ observations will be informative toward the perplexing question of NK cell education, I remain unconvinced that the pathway has been conclusively determined in this study. I agree that calcium flux, GrB accumulation, and lysosome size and localization are all correlates to NK cell effector potential, the data are insufficient to link education to managing these pathways. For instance, there is no functional data to support a link between KIR/SHP-1/ITIM (which are known to be critical for education) in manipulating the lysosomes, and this would be needed to draw the conclusion that this is a phenomenon rooted in education. The data are sufficient to conclude that effector potential is linked to the phenotypes they describe, but the finding that cytokine stimulation can phenocopy these outcomes suggests that they are instead associated with NK cell effector potential (which is known to be higher at baseline and may be controlled by a third party pathway not identified).

Author response:

We thank the reviewer for the careful evaluation of the revised manuscript and the many constructive suggestions to improve the manuscript.

We are somewhat confused by the general concern related to NK cell effector potential raised above (underlined). NK cell education is defined through differences in effector potential (eg., functional responses in a diverse set of assays) between NK cells expressing self versus non-self specific receptors:

“.....by an MHC-dependent education process described as licensing by some investigators, the NK cells that express receptors for self MHC in normal animals or humans exhibit greater responsiveness to stimulation,.....” Vivier et al., Science 2011.

State-of-the-art assays to read out NK cell education in the human include measuring CD107a and IFN γ responses in self-KIR⁺ and non-self KIR⁺ following stimulation with MHC-deficient target

cells, such as K562, or through antibody-dependent cellular cytotoxicity (ADCC), which is what we have done throughout the present study. Since education is defined through functional potential in such assays, it is necessary to relate any new phenotype (for example the one described here) to the functional response of cells with that phenotype.

No cellular or molecular mechanism of NK cell education has yet been identified, partly because there is no transcriptional signature that distinguishes educated NK cells from uneducated NK cells as shown before in mice (Gaia et al, Science Signaling 2011)⁷ and in human by us in the present study (Figure 2a). There are also very few phenotypic imprints of educated NK cells and neither of these serve as a specific marker or provide any substantial clues to the mechanisms behind NK cell education. i) Educated NK cells have higher expression of DNAM-1 in both mice and humans (Shown by us in human Enquist et al., J Immunology 2015⁸ and in mice Wagner et al, Nature Communications 2017)⁹. Two studies also reported differences in the nano-scale organization of receptors at the cell surface which correlated (*nota bene!*) with the functional potential of NK cells carrying different constellations of receptors (Guia et al., Science Signaling 2011 and Staaf et al., Science Signaling 2018).^{7, 10}

Therefore, the discovery that education, as defined by the enhanced functional potential in NK cells expressing self-specific inhibitory receptors, correlates with the level of granzyme B stored in larger lysosomal structures provides important insight into the inner workings of NK cell education. The physical compartmentalization of functional potential represents a form of molecular memory developing as a result of integrated receptor signaling under homeostasis. Based on our own experiments and evidence in other cell types, we propose that Ca²⁺ signaling from acidic stores contributes to the increased functional potential of self-KIR⁺ NK cells. Hence, we provide support for a possible mechanism that goes beyond the new and functionally relevant observation that educated NK cells carry a greater cytotoxic payload.

A direct role of KIR signaling in the granzyme B phenotype and the corresponding functional phenotype is specifically addressed in the paper by the KIR transduced YTS and NKL lines that replicate the phenotype observed in donors with different *KIR/HLA* genetics.

1. Figure S2 shows KIR3DL1 single + cells stratified by Bw4 status (A) and Bw4 subtype (B). The number of Bw4+ cells in A differs from those shown in B (presumably, the Bw4+ bar in A is simply a further stratified version of A). Why were 5 donors excluded from panel B? Even if these events were among the A24+HLA-Bx which were “too rare to be analyzed” they should be shown on panel B. Moreover, Boudreau et al., J ClinOncol 2017 demonstrated that HLA-B alleles’ impact on NK cell education superseded HLA-A*24 alleles, providing precedence that these should be included in the Iso or Thr groups (and A24 should only be composed of those with Bw6/Bw6).

Author response: We thank the reviewer for this suggestion and agree that it makes more sense to show all donors. As requested by the reviewer we include the 5 donors with rare genotypes. See revised Figure S2b below. We also cite the relevant paper on the impact of A24 vs other Bw4 alleles on page 7. Our data are entirely consistent with the notion that B alleles supersede education by A24.

“Granzyme B expression was also higher in 3DL1⁺ NK cells from donors positive for its cognate ligand HLA-Bw4 (Fig. 1c), particularly in those who possessed strong educating motifs, eg., a Bw4 allotype with isoleucine (Ile) at position 80 whereas granzyme B was lower in NK cells carrying the weak A24 motif alone (Supplementary Fig. S2a-b)^{24, 25}.”

b

Figure S2. Granzyme B expression stratified based on KIR3DL1 and HLA-Bw4 ligands. (b) Substratification showing expression of granzyme B in 3DL1sp NK cells based on the Bw4 ligands Bw4Thr80 (n=9), Bw4Iso80 (n=6), Bw4A24 (n=6), Bw4Thre80+A24 (n=4) and Bw4Iso80+A24 (n=1) were two rare to be analysed separately.

2. The authors have declined the request to stratify data by KIR3DL1 subtype due to low cell numbers. Boudreau et al (JI 2016) show that density predicts NK cell education based on its density. Perhaps an alternative approach to allele typing would be to correlate KIR3DL1 MFI with GrB density and/or to gate on the particular subpopulations of KIR3DL1 on FACS plots, where the populations can be clearly distinguished. Demonstrating a dose-response correlation would be very powerful here.

Author response: We have recently sent 300+ samples to Paul Norman at Denver for allele level typing of both KIR and HLA. This was prompted by a simple screening of granzyme B ratios between 2DL3 and 2DL1 showing that we could predict the HLA type with a high sensitivity and specificity (**Figure 1** for review purposes only). Unfortunately, 3DL1 was not included in this screening but we plan to follow up with a detailed KIR repertoire phenotyping by CyTOF. Boudreau et al (JI2016) show that education is strongest in donors with Bw4 Iso80 combined with 3DL1^{high} alleles. A separate analysis of such donors suggests that high 3DL1 expression is indeed associated with higher levels of granzyme B (**Figure 2** for review purposes only). However, as stated above, higher numbers of donors are needed to draw robust conclusions on the impact of allelic variants. The role of allelic diversity with reference to relevant literature, including the papers by Boudreau in JI and JCO is discussed on page 18.

“3DL1 allelic diversity, including the non-expressed 3DL1 null alleles, combined with Bw4 ligand polymorphism has a profound influence NK cell education.^{11, 12, 13”}

[Redacted]

Figure 1 for review only. Mean fluorescence intensity (MFI) ratio in granzyme B between 2DL2/L3/S2 and 2DL1 expressing NK cells among donors of different HLA-C phenotype. Data were generated from 345 donors. Determining HLA-C type by granzyme B ratios alone had a sensitivity of 89% and specificity of 87% as validated by subsequent genetic typing.

[Redacted]

Figure 2 for review only. Mean fluorescence intensity (MFI) of granzyme B expression in 3DL single-positive NK cells as a function of KIR3DL1 expression intensity. The analysis was restricted to donors with Bw4Iso80, previously shown to have the strongest effect on education.

3. Boudreau et al (J ClinOncol 2017) showed that the null subtype of KIR3DL1 could drive NK cell education. How were KIR3DL1-null NK cells dealt with? NK cells from donors exhibiting this allele may co-express KIR3DL1-n and therefore not be gateable.

Author response: Thanks for pointing us to this interesting observation. In the current investigation, we did not screen using intracellular stainings and could not discriminate granzyme B levels in cells co-expressing 3DL1-null alleles. The paper on 3DL1-null alleles and education is now cited and discussed on page 18. See further above in response to point 2.

4. Response to reviewer 2, 7b: While I realize that the study of cytokine-driven NK cell degranulation differs from that triggered by a missing self reaction, it is NOT beyond the scope of this paper. The authors aim to make conclusions about the differences of NK cells based on NK education. This can be accomplished in at least 2 ways: first, by comparing so uneducated NK cells from the same donors, which I acknowledge that the authors have done, and second, by proving that this has anything to do with education. I acknowledge that the responses may be different and that is the point of the query: Is the phenotype associated with education, or simply with NK cell potentiation (which would be more broadly defined)?

Author response: Please see response to the general comment. Although we agree that it would be both interesting and possible to redo the whole study with NK cells primed in different cytokines over different length of time, this is a very demanding task, since we would have to consider cytokine-induced transcription, lysosomal biogenesis, mTOR activation and subset plasticity, which all have major effects on NK cell function. Furthermore, we feel that the link to education is actually stronger when studying subsets of resting NK cell directly out of the blood using state-of-the-art methods to monitor their phenotype and functional response.

We know from work by the Walzer laboratory that IL-15-driven NK cell priming is largely mediated by mTOR activation and may (or may not) involve lysosomal modulation (given the connection between mTOR and the lysosome).¹⁴ The possible link to metabolic effect was introduced in the discussion in response to comment 7b during round 1 of this review. We do not understand how such potential downstream consequences of the structural differences in the lysosomal compartment described here, would argue against a role for the observed phenotype in the functional difference between self and non-self KIR+ NK cells at baseline.

We hope our chain of thought is clearer and that we, by responding to the additional concerns in 1-13, provide sufficient support for the proposed model to allow further scrutiny by the research community.

5. If the authors aim to show that self KIR+ NK cells are more potentiated (and more impacted by lysosomal manipulations) in figures 5 and 6, they should compare (pairwise by donor) based on treatment group between s and ns KIR.

Author response: Figure 5 and 6 show that NK cell function (CD107a and IFN) is affected by pharmacological manipulation of the lysosomal compartment in a predicted manner: Compounds that

interfere with Ca^{2+} flux from the acidic compartment leads to loss of function following receptor ligation (not following PMA/I) (Figure 5c-d), whereas inhibition of PIKfyve mimic the educated state with larger granules, more granzyme B and increased function (Figure 6g-h). The key objective of this figure is to show the significant effect of pharmacological manipulation of these pathways in NK cells. As pointed out in the first revision, a digital outcome affecting one subset but not the other is not expected from the model and from the previous work showing that education is tunable. Thus, it is not our aim to show that self-KIR⁺ NK cells are more affected. The data show that lysosomal modulation plays a role in NK cell function, in a way that is predicted by the hypothesis that the acidic compartment contributes to Ca^{2+} signaling as has been seen in T cells and other cell types.¹⁵ The link to education is the fact that self KIR⁺ NK cells show unique ultrastructural organization of the compartment, similar the once observed after pharmacological modulation of this pathway.

We have revised the result section to better reflect the key message of this figure. See page 13.

In line with concept that NK cell education is not an on/off switch but rather a continuum of functional responses, both self KIR⁺ and non-self KIR⁺ NK cells where affected by lysosomal interference.¹⁶

6. New figure 8 is a speculation of the model that links their data to education. I do not oppose the inclusion of a model system, but the link between putative activation (the disarming model has NOT itself been conclusively proven) and AKT activation is testable and, this paper's data alone cannot draw the vector from activation to PIKfyve activation. The statement that the authors have indicated where data are speculative is not sufficient to mitigate this problem (especially since it is written in a caption that requires further activation). At a MINIMUM, speculation should be indicated with dashed lines and question marks in the speculative model.

Author response: The reviewer raises a fair point that is well taken. We have followed the advice and dashed arrows describe steps that were not specifically addressed in the current paper, including the pathways leading to activation of PIKfyve. The term disarming has been deleted from the model. While we appreciate that the study would be even stronger by providing detailed insights into the complete signaling pathway upstream of TRPML1, we believe the study does provide sufficient new knowledge as is to advance the field. Indeed, the potential role of inter-organelle communication is very much in the spotlight in the context of immune cell activation.¹⁵ Thus, we expect considerable interest from immunologists, cell biologists and cell signalers alike.

7. The MFI of GrB in figure 6 implies that the cells have been selected for self-KIR⁺, but it is not explicitly stated that this is the case. Does inhibition of PIKfyve lead to the same phenotype in educated and uneducated cells? If it does, this may assist in developing the mechanistic conclusion that lysosome size and GrB loading is a direct correlate to NK cell potentiation.

Author response: Thanks for this suggestion. The original figure was based on analysis of bulk CD56^{dim} NK cells from 7 donors. In five out of these, we could distinguish self-KIR⁺ NK cells from non-self KIR⁺ NK cells. Indeed, PIKfyve inhibition leads to increased levels of granzyme B in both subsets, strengthening the link between the granzyme B phenotype and NK cell function. This new analysis is shown in new Figure 6b. Text on page 14 has been revised accordingly.

Figure 6. Enlarging the secretory lysosomes leads to enhanced NK cell functionality. (b) Intracellular granzyme B expression in self-KIR⁺ and non-self-KIR⁺ NK cells following overnight incubation with the indicated PIKfyve inhibitor assessed by flow cytometry (n=5).

8. If paired t-tests are used in Figure 7, the authors should also include some indicator of which samples are the pairs (either by specific colors or joining lines). As presented, the data are unconvincing of differences. I note that there are different numbers of samples in the panels and throughout the manuscript. Were power calculations undertaken to determine the optimal number of replicates?

Author response: Paired t-tests were performed in panels b, c, e, i and j. We have revised all the figures with paired comparisons in Figure 7 with connecting lines. We have also included information about the number of experiments and donors in the legend. The number of replicates and donors vary between experimental series and the choice of statistical method was based on the type of comparison made and the distribution of the data as outlined in the method section. No specific power calculations were made for the individual experimental series. We have made every effort to analyse sufficient donors for robust conclusions to be drawn. For example, during revision we analysed a set of 49 additional donors to address the accumulation of perforin.

9. If activating stimulation is required, what is its source in the transfection models shown in Figure 1h? That cells exhibit an education-like phenotype with respect to GrB after this transfection supports education by inhibitory interactions with self MHC.

Author response: YTS cells are cultured in the absence of cytokines whereas NKL require IL-2 for survival. This information is now provided in the methods section on page 24. We agree that these experiments complement the snapshot analysis in donors with different *KIR/HLA* genetics and provide support for a direct role of the inhibitory KIR in the accumulation of granzyme B.

10. The color bar scale is missing from Figure 1b in the manuscript (but not rebuttal).

Author response: In response to concerns raised about the validity of the SNE plot we have decided to delete this panel from the manuscript.

11. Separation of KIR2DL2 single positive cells is still unclear to me. The combination of antibodies for KIR2DL3 and KIR2DL2/L3/S2 enables isolation of KIR2DL3 from this group, but how is KIR2DS2 excluded?

Author response: This point was also raised by reviewer 1. The reviewer is right that we cannot discriminate 2DS2 from 2DL2 in these stainings. We have corrected the figure, legend and the text.

12. Many educated NK cells exhibit CD57. Why were cells exhibiting this marker excluded? At a minimum, I recommend including a description of the data for CD57⁺ NK cells.

Author response: This is an important point. In the revised manuscript, we have re-analysed the granzyme B levels in NKG2A⁻CD57⁺ self KIR⁺ versus non-self-KIR⁺ NK cells (Supplementary Figure

9e, see below). As shown in this figure, the effect of self-KIR expression is seen also in CD57⁺ NK cells. These data are discussed on page 11.

“Importantly, the accumulation of effector molecules in educated self-KIR⁺ NK cells was observed also in more differentiated NKG2A-CD57⁺ NK cells (Supplementary Fig. S9e).”

Supplementary Figure 9e. Expression of granzyme B in NKG2A-CD57⁺ NK cells expressing the indicated KIR in C1/C1 (n=16), C1/C2 (n=18) and C2/C2 (n=13) donors.

13. In figures 2a and b, the authors sort KIR⁺ NK cells (but not self-KIR⁺) and draw the conclusion that self-KIR⁺ NK cells develop according to the transcriptional program they describe. This conclusion cannot be made without comparing to non-self KIR⁺ NK Cells AND separating the self from the non-self populations.

Author response: We have indeed sorted NKG2A-CD57-Self KIR⁺ and NKG2A-CD57-NonSelf KIR⁺ NK cells and show for the first time in human that these have identical transcriptomes (Figure 2a and b). This information was unintentionally lost in the legend of the revised manuscript. We apologize for this mistake that has now been corrected. It is explicitly stated in the result section and indicated in the figure. We hope this comes across more clearly in the revised ms.

In parallel, we have sorted NK cells at different stages of differentiation and show that acquisition of KIR is linked to acquisition of an effector program (Figure 2b and Supplementary Figure S5). Given that there is no difference in the transcriptomes between self KIR⁺ and nonself KIR⁺ NK cells (2a and b), the conclusion drawn from the data shown in Supplementary Figure S5 concerning acquisition of effector programs upon KIR acquisition is valid for both non-self KIR⁺ and self-KIR⁺ NK cells. Our interpretation of these results is that NK cell differentiation provides the template for transcription of effector molecules (including lysosomal biogenesis) that is then modulated by self KIR-HLA interactions.

References

1. Romagne, F. *et al.* Preclinical characterization of 1-7F9, a novel human anti-KIR receptor therapeutic antibody that augments natural killer-mediated killing of tumor cells. *Blood* **114**, 2667-2677 (2009).
2. Liu, L.L. *et al.* Critical Role of CD2 Co-stimulation in Adaptive Natural Killer Cell Responses Revealed in NKG2C-Deficient Humans. *Cell Rep* **15**, 1088-1099 (2016).
3. Beziat, V. *et al.* NK cell responses to cytomegalovirus infection lead to stable imprints in the human KIR repertoire and involve activating KIRs. *Blood* **121**, 2678-2688 (2013).
4. Bjorkstrom, N.K. *et al.* Analysis of the KIR repertoire in human NK cells by flow cytometry. *Methods Mol Biol* **612**, 353-364 (2010).
5. Andersson, S., Malmberg, J.A. & Malmberg, K.J. Tolerant and diverse natural killer cell repertoires in the absence of selection. *Exp Cell Res* **316**, 1309-1315 (2010).

6. Fauriat, C. *et al.* Estimation of the size of the alloreactive NK cell repertoire: studies in individuals homozygous for the group A KIR haplotype. *J Immunol* **181**, 6010-6019 (2008).
7. Guia, S. *et al.* Confinement of activating receptors at the plasma membrane controls natural killer cell tolerance. *Science signaling* **4**, ra21 (2011).
8. Enqvist, M. *et al.* Coordinated expression of DNAM-1 and LFA-1 in educated NK cells. *J Immunol* **194**, 4518-4527 (2015).
9. Wagner, A.K. *et al.* Expression of CD226 is associated to but not required for NK cell education. *Nat Commun* **8**, 15627 (2017).
10. Staaf, E. *et al.* Educated natural killer cells show dynamic movement of the activating receptor NKp46 and confinement of the inhibitory receptor Ly49A. *Science signaling* **11** (2018).
11. Boudreau, J.E., Mulrooney, T.J., Le Luque, J.B., Barker, E. & Hsu, K.C. KIR3DL1 and HLA-B Density and Binding Calibrate NK Education and Response to HIV. *J Immunol* **196**, 3398-3410 (2016).
12. Forlenza, C.J. *et al.* KIR3DL1 Allelic Polymorphism and HLA-B Epitopes Modulate Response to Anti-GD2 Monoclonal Antibody in Patients With Neuroblastoma. *J Clin Oncol* **34**, 2443-2451 (2016).
13. O'Connor, G.M. *et al.* Mutational and structural analysis of KIR3DL1 reveals a lineage-defining allotypic dimorphism that impacts both HLA and peptide sensitivity. *J Immunol* **192**, 2875-2884 (2014).
14. Bjorkstrom, N.K. *et al.* Expression patterns of NKG2A, KIR, and CD57 define a process of CD56dim NK-cell differentiation uncoupled from NK-cell education. *Blood* **116**, 3853-3864 (2010).
15. Davis, L.C. *et al.* NAADP activates two-pore channels on T cell cytolytic granules to stimulate exocytosis and killing. *Curr Biol* **22**, 2331-2337 (2012).
16. Brodin P., Karre, K. & Hoglund, P. NK cell education: not an on-off switch but a tunable rheostat. *Trends Immunol* **30**, 143-149 (2009).

REVIEWERS' COMMENTS:

Reviewer #1 (Remarks to the Author):

The paper has been properly revised. In my opinion, it is now suitable for publication in Nature Communications.

Reviewer #2 (Remarks to the Author):

I thank the authors for considering our comments and taking time to respond, including the addition of more data and thoughtful analysis. I am concerned, however, that the definition of NK "education" is unclear and could lead to an over interpretation of conclusions. What the paper is showing is potentiation of NK cells, driven by a phenotype of increased dense-core granules with centromeric localization, a process that can be driven through PIKFYVE etc. The link to education a subset of "potentiation" that concerns programming and reprogramming of NK cells through inhibitory interactions remains unsupported by the data presented here.

The authors present only one panel of a figure to show that introduction of a self-sensitive KIR receptor alters the composition of an NK cell line and state in their rebuttal that they cannot prove inhibition or even replicate the experiment because the cell line is not currently available in their laboratory. Cell lines are notoriously independent of NK cell education and easily triggered, so I caution about making conclusions solely based on transfected cell line data (especially on a single experiment). Data with additional cell lines and primary cells (for instance, through knockdown or introduction of sKIR) would be necessary to substantiate these claims. Signaling through sKIR (including SHIP) is necessary for education, so would introduction of a KIR with a mutated ITIM (or a SHIP knockout cell) be unable to generate the "potentiated" phenotype?

It will be quintessential to disentangle education from potentiation – although potentiation may be the mechanism through which education is made possible, the data that cytokine stimulation or differentiation (shown as CD57+) are associated with increases in GrB MFI supports a correlation between potentiation and GrB accumulation (granule localization and size are not shown in those experiments), that may be independent of education. In other words, the pathways driving lysosomal size/granule regulation to KIR have not been demonstrated – these are coincident findings until this pathway is identified. Education is predicated on sensitivity to "self" HLA, not simply a programming of NK cells for effector function (the latter is a more broad definition). I agree that there is a need to define markers for NK cell education; none, other than sKIR expression, are 100% effective. My problem with this is not that there is a correlation between education and a phenotype (which is indeed what Guia and Staaf demonstrated), but with the conclusion that it is THE MECHANISM by which education is conferred. I would be less critical if the title were "Modulation of secretory lysosomes for accumulation of GrB is associated with enhanced effector potential and NK cell education".

The text indicates that 64 donors were examined in panel 1b. Why aren't all of the donors shown? Also, since the intent of this and the other panels is to illustrate differences in education by comparing donors' subpopulations based on their KIR ligands, panel A should similarly be shown with adjoining lines and analyzed by paired samples t tests.

REVIEWERS' COMMENTS:

Reviewer #1 (Remarks to the Author):

The paper has been properly revised. In my opinion, it is now suitable for publication in Nature Communications.

Reviewer #2 (Remarks to the Author):

I thank the authors for considering our comments and taking time to respond, including the addition of more data and thoughtful analysis. I am concerned, however, that the definition of NK “education” is unclear and could lead to an over interpretation of conclusions. What the paper is showing is potentiation of NK cells, driven by a phenotype of increased dense-core granules with centromeric localization, a process that can be driven through PIKFYVE etc. The link to education a subset of “potentiation” that concerns programming and reprogramming of NK cells through inhibitory interactions remains unsupported by the data presented here.

Author response. We wish to thank the reviewer for the careful scrutiny of our work. We understand the reviewer’s point concerning functional potentiation better after reading this third round of reviews. We comment on this further below in relation to the work dealing with pharmacological modulation of the lysosomal compartment to phenocopy education. However, we disagree with the generalized statement concerning an unsupported link between education and lysosomal modulation. The phenotypic definition of education used in this ms to prove a relationship between education and lysosomal modulation is the exact same definition proposed by the reviewer below, namely the expression of self versus non-self KIR by NK cells at a given stage of differentiation.

The key observation in this manuscript is that expression of self KIR is associated with morphological changes in the cell (dense-core secretory lysosomes which are converged to the centrosome). This morphological change, noted in confocal microscopy and immuno-EM, is reflected in accumulation of effector molecules such as granzyme B and perforin and thus has direct consequences on the cytotoxic capacity of the cell. Our data support the notion that functional tuning through self-specific receptors occurs after their acquisition of an effector program during NK cell differentiation. The latter should not be confused with a direct role for differentiation (for example reflected in CD57 expression) or cytokines (through increased translation) in causing the difference in the lysosomal compartment between self and non-self KIR+ NK cell subsets. Indeed, we have made several efforts to exclude a role for transcriptional differences associated with differentiation as an underlying factor explaining the observed morphological differences between self-KIR+ and non-self KIR+ NK cells.

The authors present only one panel of a figure to show that introduction of a self-sensitive KIR receptor alters the composition of an NK cell line and state in their rebuttal that they cannot prove inhibition or even replicate the experiment because the cell line is not currently available in their laboratory. Cell lines are notoriously independent of NK cell education and easily triggered, so I caution about making conclusions solely based on transfected cell line data (especially on a single experiment). Data with additional cell lines and primary cells (for instance, through knockdown or introduction of sKIR) would be necessary to substantiate these claims. Signaling through sKIR

(including SHIP) is necessary for education, so would introduction of a KIR with a mutated ITIM (or a SHIP knockout cell) be unable to generate the “potentiated” phenotype?

Author response: We provide four independent and replicated series of experiments using two NK cell lines and do think these may prove useful to dissect the mechanism further in future experiments. We regret not being able to provide more data from the cell lines for the current revision. The strategies proposed above are all relevant and we are currently engineering iPSC-derived NK cells to explore these points further. However, it is worth pointing out that all primary cultures of NK cells (and NKL cells) require addition of cytokines for survival and are therefore suboptimal for studies of NK cell education (discussed at length in previous rebuttal letters). Although the analysis of resting NK cell repertoires represent snapshots at any given point in time, the stochastic expression of self and non-self KIR across a large cohort of donors (100+) with different HLA backgrounds provide a unique natural in vivo gene-silencing system to study the impact of specific KIR-HLA constellations.

It will be quintessential to disentangle education from potentiation – although potentiation may be the mechanism through which education is made possible, the data that cytokine stimulation or differentiation (shown as CD57+) are associated with increases in GrB MFI supports a correlation between potentiation and GrB accumulation (granule localization and size are not shown in those experiments), that may be independent of education. In other words, the pathways driving lysosomal size/granule regulation to KIR have not been demonstrated – these are coincident findings until this pathway is identified. Education is predicated on sensitivity to “self” HLA, not simply a programming of NK cells for effector function (the latter is a more broad definition). I agree that there is a need to define markers for NK cell education; none, other than sKIR expression, are 100% effective.

My problem with this is not that there is a correlation between education and a phenotype (which is indeed what Guia and Staaf demonstrated), but with the conclusion that it is THE MECHANISM by which education is conferred. I would be less critical if the title were “Modulation of secretory lysosomes for accumulation of GrB is associated with enhanced effector potential and NK cell education”.

Author response: We believe the observation that educated self-KIR+ NK cells display altered morphology and load more granzyme B is more than a marker and provide important clues to the inner workings of education. The data represent the first intracellular phenotype of the educated state and clearly put lysosomal modulation in the spotlight. The phenotype we describe is immediately relevant for the functional potential of the cell since it is in the core cytolytic machinery itself.

In the manuscript we have also begun to address possible mechanisms operating upstream of the lysosomal phenotype. Using pharmacological interventions and genetic silencing, we have explored the PIKfyve/TRPML1 axis, known to be relevant for lysosomal modulation, and are able to phenocopy the increased functional potential associated with education by interfering with this pathway. We are not proposing a new role for inhibitory KIR. The model simply suggests that inhibitory KIR shut down any upstream activation during homeostasis. This piece of the puzzle clearly requires more work. The reviewer is right that it cannot be excluded that these agents/silencing experiments boost NK cell function in a way that is independent of NK cell education, although the data fit the predictions made a priori based on the observed lysosomal phenotype in educated self-KIR+ NK cells. The key challenge is to demonstrate how unopposed, weakly agonistic, activating signals directly activate the PIKfyve/TRPML1 axis or some other pathway that shape the lysosomal compartment. This work lies ahead of us and hopefully the community as a whole will find an interest in exploring these pathways.

Using state-of-the-art tools, we have explored how an enlarged acidic compartment may contribute to the generally increased function of self-KIR+ NK cells. This is a very important area to explore further since the mere accumulation of granzyme B alone cannot explain the propensity of self-KIR+ NK cells to respond better to all stimulations and produce more IFN gamma. We believe our data, in line with previous observations in Jurkat T cells, support a role for the acidic compartment in receptor signaling. However, as illustrated in Figure 8, it remains unclear how the release/oscillation of bound Ca in the acidic compartment contribute to this response.

We are confident that the knowns and unknowns of the model are clearly described in the manuscript, including which pieces that are supported by our data, the literature and which pieces that remains to resolve.

The title has been shortened to reflect the key message of the paper. “Remodeling of Secretory Lysosomes During Education Tunes Functional Potential in NK Cells”.

The text indicates that 64 donors were examined in panel 1b. Why aren't all of the donors shown? Also, since the intent of this and the other panels is to illustrate differences in education by comparing donors' subpopulations based on their KIR ligands, panel A should similarly be shown with adjoining lines and analyzed by paired samples t tests.

Author response: We thank the reviewer for noting this mistake. The figure only included 55 of the 64 donors. We have gone through the source data and updated the figure with missing data from two C2/C2 donors and two C1/C1 donors. Statistics remain unchanged. The total number of donors in this graph is 59, C2/C2 (n=12), C1/C2 (n=26), C1/C1 (n=21). Among the 64 donors included in the study four of the C2/C2 and one of the C1/C1 donors had too few 2DL1+ NK cells (below 100) and were therefore excluded. Legends have been amended. The number of donors in panel A and B have been corrected along with the revision of panel A and revised statistics for the data in panel A as suggested.